# Upper Jurassic carbonate buildups in the Miechów Trough, Southern Poland – insights from seismic data interpretation

Łukasz Słonka[1], Piotr Krzywiec[1]

[1]Institute of Geological Sciences, Polish Academy of Sciences (IGS PAS), Twarda Street 51/55, 00-818 Warsaw, Poland

*Correspondence to*: Łukasz Słonka (lukasz.slonka@twarda.pan.pl)

**Abstract.** The geometry and internal architecture of the Upper Jurassic carbonate depositional system in the epicontinental basin of western and central Europe, and within the northern margin of the Tethyan shelf are hitherto only partly recognised, especially in areas with thick Cretaceous and younger cover such as the Miechów Trough. In such areas, seismic data are indispensable for analysis of a carbonate depositional system, in particular for identification of the carbonate buildups and the

enveloping strata. The study area is located in the central part of the Miechów Trough that in the Late Jurassic was situated within the transition zone between the Polish part of western and central European epicontinental basin and the Tethys Ocean. This paper presents the results of interpretation of 2D seismic data calibrated by deep wells that document the presence of large Upper Jurassic carbonate buildups. The lateral extent of particular structures is in the range of 400–1000 m, and their heights are in range of 150–250 m. Interpretation of seismic data revealed that the depositional architecture of the subsurface

Upper Jurassic succession in the Miechów Trough is characterised by the presence of large carbonate buildups surrounded by basinal (bedded) limestone-marly deposits. These observations are compatible with depositional characteristics of well-recognised Upper Jurassic carbonate sediments that crop out in the adjacent Kraków-Częstochowa Upland. The presented study provides new information about carbonate open shelf sedimentation within the transition zone in the Late Jurassic, which proves the existence of much more extensive system of organic buildups which flourished in this part of the basin. Obtained

results, due to high quality of available seismic data, provide also an excellent generic reference point for seismic studies of carbonate buildups from other basins and of different ages.

## 1 Introduction

Carbonate buildups display considerable vertical accretion to adapt to a gradual relative sea level rise (e.g. Kendall and Schlager, 1981; Read, 1985; Sarg, 1988; Handford and Loucks, 1993; Schlager, 2005). The term "carbonate buildup", often

used in seismic stratigraphic studies, refers to all "carbonate deposits that form positive bathymetric features" (Bubb and Hatlelid, 1977). Seismic data proved to be very useful for the identification of carbonate buildups, because they can clearly show the differences in depositional characteristics between the buildup and the enveloping strata. Carbonate beds are often related to relatively high reflectivity of seismic data. Lateral and vertical variations of this reflectivity (including amplitude and frequency characteristics, continuity of seismic horizons, etc.), and related considerable differences in seismic velocities

of particular rock packages are related to different lithologies within the carbonate buildups and surrounding deposits (see Fontaine et al., 1987; Macurda, 1997, for overview of seismic facies analysis of carbonate rocks).

Seismic expression of carbonate buildups may be rather diverse (Fig. 1). Classical interpretation, established during the period of intense development of seismic stratigraphy in the late 1970's, assume several recognition criteria, such as the (1) mound-shaped reflection configuration pattern, (2) lateral seismic facies changes between the buildups and enveloping beds, (3) reflections from the edges of buildups including hyperbolic diffractions, (4) onlap of overlying strata, (5) drape effects over the buildups, (6) the velocity pull-up anomalies etc. (e.g. Bubb and Hatlelid, 1977; Veeken and Van Moerkerken, 2013; Burgess et al., 2013). Also differential compaction (manifested by so-called compaction sag) might indicate the presence of a carbonate buildup on seismic data.

Numerous papers dealing with various aspects of seismic interpretation of carbonate buildups of different ages have been published over the years, concerning various sedimentary basins such as for example the Great Bahama Bank (e.g. Eberli et al., 2004), Maldives (e.g. Belopolsky and Droxler, 2004), South Oman (e.g. Borgomano et al., 2004), Northern Australia (e.g. Isern et al., 2004; Rosleff-Soerensen et al., 2012; Saqab and Bourget, 2016; Van Tuyl et al., 2018, 2019), Black Sea region (e.g. Afanasenkov et al., 2007; Guo et al., 2011), offshore southern Norway (e.g. Philips et al., 2019) the Barents Sea, Norway (e.g. Blendinger et al., 1997; Elvebakk et al., 2002; Colpaert et al., 2007; Rafaelsen et al., 2008; Di Lucia et al., 2017; Sayago et al., 2018), Philippines (e.g. Grötsch and Mercadier, 1999; Neuhaus et al., 2004; Fournier et al., 2004, 2005; Fournier and Borgomano, 2007), South China Sea (e.g. Wu et al., 2009; Yubo et al., 2011; Chang et al., 2017), offshore Indonesia and Malaysia (e.g. Epting, 1989; Kusumastuti et al., 2002; Zampetti et al., 2003, 2004; Bachtel et al., 2004; Posamentier et al., 2010; Koša, 2015), or Indus Basin (Shahzad et al., 2018, 2019). In contrast, relatively few seismic examples of Upper Jurassic carbonate buildups from classic, geologically well recognised western and central European northern Tethyan shelf and surrounding region have been published to date (e.g. Ellis et al., 1990; Zimmer and Wessely, 1996; Adámek, 2005; Wessely, 2006; Bunes et al., 2010; Hartmann et al., 2012; Lüschen et al., 2014; Fig. 2). In Poland, also only a few papers on seismic interpretation of the Upper Jurassic carbonate buildups have been published so far, usually in local journals and focused mostly on exploration-related problems (Gliniak et al., 2000, 2001, 2005; Gliniak and Urbaniec, 2001, 2005; Misiarz, 2003; Misiarz et al., 2004; Jędrzejowska-Tyczkowska et al., 2005, 2006; Myśliwiec et al., 2006). Recently, several new buildups have been identified and interpreted using seismic data (Urbaniec, 2019). However those results represent more southern part of the basin, located beneath the Miocene of the Carpathian Foredeep Basin (about 50 km to the south from the study area).

This study fills this gap and provides a well-documented example of a system of carbonate buildups developed in the south-eastern segment of the transition zone between the European Late Jurassic epicontinental basin and the Tethys Ocean. Results presented in this paper could also be used as a more universal reference point for seismic studies of carbonate depositional systems, in particular of carbonate buildups and surrounding deposits, of different ages and from different sedimentary basins.

The study area is located in the central part of the Miechów Trough, southern Poland, approximately 50 km north-east from Kraków, in the vicinity of the town of Pińczów (Fig. 3). In this area, the geometry and depositional architecture of the Upper Jurassic carbonate succession is relatively poorly recognised in comparison to adjacent parts of the basin in Poland and central

and western Europe. This is mostly due to the fact that the Jurassic succession is covered by relatively thick Cretaceous and younger deposits so previous studies were based almost entirely on data from deep research wells (cf. Złonkiewicz, 2006, 2009).

In 2011, the Upper Jurassic carbonate succession in the study area was drilled by two exploratory wells, Chopin-1 and Belvedere-1. These two wells, together with 3 archive wells located in this area, were used to calibrate relatively dense coverage of 2D seismic reflection profiles. Synthetic seismograms were used to precisely tie wells to seismic profiles, and seismo-stratigraphic approach was used to analyse depositional architecture of the Upper Jurassic carbonate system, including carbonate buildups and surrounding deposits.

## 2 Geological Setting

### 2.1 The Permian–Mesozoic Polish Basin: an overview

The study area is located within the central part of the Miechów Trough (Fig. 3), which forms the south-eastern part of the Szczecin–Łódź–Miechów Synclinorium (Żelaźniewicz et al., 2011), that was formed during the Late Cretaceous–Paleogene inversion of the Permian–Mesozoic Polish Basin.

The Permian–Mesozoic Polish Basin formed the easternmost part of a system of epicontinental basins in western and central Europe (Ziegler, 1990; Scheck-Wenderoth et al., 2008; Pharaoh et al., 2010). Its most subsiding axial part – the Mid-Polish Trough – evolved along the NW- to SE-trending Teisseyre-Tornquist–Zone (see Mazur et al., 2015 for a recent summary and further references). The south-eastern part of the Polish Basin extended into the transition zone towards the Tethyan domain, characterised by limited Permian and Triassic sedimentation. Since the Jurassic, the thickness and depositional pattern in this part of the basin was affected by tectonic processes acting within the Polish Basin, and by increased regional subsidence in the Tethyan domain (e.g. Kutek and Głazek, 1972; Pożaryski and Żytko, 1981; Feldman-Olszewska, 1997a, 1997b; Marek and Pajchlowa, 1997; Dadlez et al., 1998; Kutek, 2001; Gutowski et al., 2005; Gutowski and Koyi, 2007; Krzywiec et al., 2009).

The Polish Basin was inverted in the Late Cretaceous–Paleogene (e.g. Dadlez et al., 1995; Krzywiec, 2002, 2009; Resak et al., 2008; Krzywiec et al., 2009, 2018). This basin inversion was associated with major uplift and erosion of the axial part of the basin (i.e. the Mid-Polish Trough), which was transformed into a regional anticlinal structure – the Mid-Polish Swell (Mid-Polish Anticlinorium; cf. Pożaryski and Brochwicz-Lewiński, 1978, 1979; Żelaźniewicz et al., 2011). Due to inversion-related formation of the Mid-Polish Swell, two regional synclinoria were formed along both its flanks, including the south-western Szczecin–Łódź–Miechów Synclinorium, where the Miechów Trough is located (see e.g. Dadlez et al., 2000; Fig. 3).

### 2.2 Late Jurassic basin in S Poland

The Late Jurassic basin in Poland formed the eastern part of extensive shallow epicontinental basin that extended from the United Kingdom, across the Netherlands and Germany, into Poland and farther into the east (Fig. 2; Ziegler, 1990; Pieńkowski

et al., 2008; Lott et al., 2010). Throughout much of the Jurassic, the basin was connected to the Tethys Ocean from the south (Lott et al., 2010, see Pieńkowski et al., 2008 for detailed overview and further references). The Late Jurassic was a time of extensive development of carbonate buildups in the Tethyan domain and its margins (e.g. Leinfelder et al., 1994, 2002; Matyszkiewicz, 1997a; Krajewski and Schlagintweit, 2018).

In the Late Jurassic, the south-east (peri-Carpathian) segment of the basin was part of the European shelf adjacent to the
Tethys Ocean from the north (cf. Ziegler, 1990; Golonka, 2004; Golonka et al., 2000; Gutowski et al., 2005, 2006; Pieńkowski et al., 2008). The main factors that directly or indirectly controlled Late Jurassic sedimentation within the northern Tethyan shelf in southern Poland included sea-level and climate change, and diversified subsidence triggered by reactivation of older basement faults (e.g. Kutek, 1994; Gutowski et al., 2005; Matyszkiewicz et al., 2012, 2015a, 2016).

The Oxfordian and lower Kimmeridgian succession within the Polish part of the northern Tethyan shelf margin is
commonly interpreted as a carbonate ramp or an open shelf deposits (e.g. Matyja et al., 1989; Kutek, 1994; Gutowski et al., 2005; Matyja, 2009; Krajewski et al., 2011; Fig. 4a). These deposits, sometimes termed the sponge megafacies, are bulit of sponges and microbialites, and are present within the entire European part of the northern Tethyan shelf margin (Gwinner, 1971; Matyja, 1977; Matyja and Pisera, 1991; Matyja and Wierzbowski, 1995, 1996, 2006; cf. Matyszkiewicz, 1997a; Gutowski et al., 2005, 2006).

Widespread carbonate sedimentation took place in the Oxfordian (upper Transversarium–Bifurcatus and Planula Zones), when diverse reef facies developed (Fig. 4a; e.g. Matyszkiewicz et al., 2012, 2015b, 2016; Krajewski et al., 2016, 2018). Several authors claim that development of carbonate platforms in this part of Europe may have been connected to the Middle Oxfordian (Transversarium Zone) climate warming (Krajewski et al., 2017; see Leinfelder et al., 1996; Matyszkiewicz, 1997a; Olivier et al., 2011; Wierzbowski, 2015). The Upper Jurassic carbonate buildups in southern Poland display a large diversity
of reef types, from siliceous sponge mounds to microbial-sponge buildups and coral reefs, as all of these types were commonly found in Europe where reefs were most widespread in the Late Jurassic (Kiessling et al., 1999; cf. Leinfelder et al., 1996; Gliniak et al., 2005; Matyszkiewicz et al., 2012; Krajewski et al., 2018). Outside of Europe reefs occurred less commonly in Late Jurassic, and they represented mainly coral-dominated reefs and biostromes (Kiessling et al., 1999). Common carbonate buildup types that can be recognized from the seismic data in Poland are bioherms (e.g. Gliniak and Urbaniec, 2001, 2005;
Gliniak et al., 2005). Worldwide, these organic structures can be found in all latitudes between 45°S and 52°N (Kiessling et al., 1999); in southern Poland they often developed as large microbial-sponge biohermal complexes (e.g. Matyja and Wierzbowski, 2006).

The Callovian to Lower Kimmeridgian (up to Hypselocyclum Zone) deposits of the Polish Basin have been subdivided by Kutek (1994) into two intervals related to distinct stages of tectono-sedimentary evolution (Krajewski et al., 2017; cf. Kutek,
1994). The first one embraces Callovian–Oxfordian, including the Planula Zone; it is commonly limited to the Upper Oxfordian in the Sub-Mediterranean subdivisions (e.g. Krajewski et al., 2017), whereas the second interval encompass the Lower Kimmeridgian (Platynota–Hypselocyclum zones). Both intervals are separated by the so-called Lowermost Marly Horizon, included in the Lower Platynota Zone, which plays an important role of a regional isochronous marker in stratigraphic

correlations of the Upper Jurassic in central and southern Poland (Kutek, 1968, 1994). Between those two intervals, significant

facies changes occurred (e.g. Kutek, 1994; Matyszkiewicz, 1996; Krajewski et al., 2017). They are expressed by: (1) the disappearance of the Oxfordian organic buildups, (2) platform drowning in the Lower Platynota Zone linked with development of marly facies, and (3), occurrence of gravity-flow deposits (e.g. Krajewski et al., 2017).

### 2.3 The Upper Jurassic succession in the Miechów Trough

The Upper Jurassic succession of the Miechów Trough is almost entirely covered by Cretaceous deposits, represented by the Albian–Lower Maastrichtian (Fig. 3; e.g. Jurkowska, 2016), and, in its south-eastern part, by the Miocene deposits of the Carpathian foredeep basin (e.g. Pożaryski, 1977; Żytko et al., 1988; Krzywiec, 2001). From the south-west, the Miechów Trough borders with the Kraków-Częstochowa Upland, and from the north-east, within the Holy Cross segment of the Mid-Polish Anticlinorium (Fig. 3; cf. Pożaryski, 1974; Żelaźniewicz et al., 2011).

The Upper Jurassic carbonate deposits outcropping along the flanks of the Miechów Trough, have been extensively studied for many decades (e.g. Dżułyński, 1952; Kutek, 1968, 1969; Matyja, 1977; Matyja and Tarkowski, 1981; Trammer, 1982, 1985, 1989; Matyszkiewicz, 1989, 1993, 1996, 1997b, 1999, 2001; Matyszkiewicz and Felisiak, 1992; Matyja and Wierzbowski, 1996, 2006; Matyja et al., 1989, 2006; Matyszkiewicz et al., 2006, 2012, 2015a, 2016; Krajewski et al., 2011, 2016, 2017, 2018). Numerous studies dealing with detailed aspects of the Upper Jurassic stratigraphy and sedimentology have

been carried out also in the more south-eastern part of the Miechów Trough, including its extension towards the Carpathian foredeep, south from the Wisła river (cf. Fig. 3; Morycowa and Moryc, 1976, 2011; Golonka, 1978; Gliniak et al., 2004; Gutowski et al., 2005, 2007; Matyja and Barski, 2007; Matyja, 2009; Olszewska et al., 2012).

The Upper Jurassic succession in the Miechów Trough is represented by various carbonate ramp-type platform facies (e.g. Kutek 1968, 1969; Matyja et al., 1989, 2006; Gutowski et al., 2005, 2006; Matyja, 2009; Złonkiewicz, 2009; Krajewski et al.,

2017; Fig. 4a). According to Złonkiewicz (2009), the Callovian and Upper Jurassic deposits in the Miechów Trough genetically resemble those from the south-western margin of the Holy Cross Mts. (cf. Matyja et al., 1989), which prompted him to adopt almost the same lithostratigraphic correlation scheme (Złonkiewicz, 2009).

During the Late Jurassic, the study area was located on the northern, passive margin of the Tethys Ocean (e.g. Matyja and Wierzbowski, 1995; Golonka, 2004; Matyja, 2009). Sequence stratigraphic scheme for this part of the basin, together with

regional correlation of main depositional systems, was proposed by Gutowski et al. (2005). This scheme can be generally correlated with the main Oxfordian–Kimmeridgian lithological units in the study area (Złonkiewicz, 2009; Fig. 4a and b). These key Upper Jurassic units include the Morawica Limestone Member, Siedlce Limestone Member and Massive Limestone Member (Matyja et al., 1989; Złonkiewicz, 2009; Krajewski et al., 2017; Fig. 4b). Above those carbonate members, deeper-water marly facies are present (Kutek, 1968). They are covered by deposits of the Early Kimmeridgian shallow-water carbonate

platform, represented by various oolitic-platy facies (Fig. 4b; see Złonkiewicz, 2009 for more details).

For the south-easternmost part of the Miechów Trough, located beneath the Miocene cover of the Carpathian Foredeep basin, detailed subdivision of the Upper Jurassic deposits has been recently proposed using biostratigraphic data (Matyja and Barski, 2007; Barski and Matyja, 2008; Matyja, 2009). According to this stratigraphic scheme, a complete Oxfordian–Valanginian succession is present in the most south-eastern part of the basin, with significantly lower than previously assumed thickness of the Oxfordian–Kimmeridgian deposits, and much more extensive, in comparison to other areas of Poland, stratigraphic range of the sponge megafacies, reaching up to the lower Tithonian (Matyja, 2009). Further micropaleonotlogical investigations allowed for stratigraphical reassessment of the Upper Jurassic strata beneath the central part of the Carpathian Foredeep basin, as well as for the regional correlations towards the south-western Ukraine (Olszewska et al. 2012). These findings could possibly also be applied in the future to stratigraphy of the Upper Jurassic succession of the area described in this paper, although this would require extensive studies based on core material that is not currently available.

## 3 Data and methods

### 3.1 Well data

Well calibration for seismic data in this study was provided by the Chopin-1, Belvedere-1, Michałów-3, Węchadłów-1 and Lipówka-1 wells (Fig. 5). Two of these wells, Chopin-1 and Belvedere-1, have been drilled in 2011 by the San Leon Energy company (SLE); the other 3 wells were drilled in mid-1960's. Therefore, the suitability of well data for detailed seismic analysis was diverse. Both the SLE wells have a wide spectrum of modern well log data, including gamma-ray, resistivity, neutron porosity, sonic velocity and density logs, as well as mud-logging; they however haven't been cored and lithological descriptions are based on cuttings. The data used from three legacy wells included gamma-ray, resistivity and sonic logs. All the logs were available as standard LAS files and were loaded into the databased used in this study.

Stratigraphic information for the Upper Jurassic succession substantially differs between older wells and two newer SLE wells. In the legacy wells, the Upper Jurassic interval was subdivided into Oxfordian, Rauracian and Astartian (Mikucka-Reguła, 1968; Urban and Wandas, 1968; see also Kutek, 1965). Since late 1960's– early 1970's, Rauracian and Astartian have been incorporated into the upper Oxfordian (e.g. Morycowa and Moryc, 1976). On the other hand, the Upper Jurassic interval in the SLE wells Chopin-1 and Belvedere-1 was subdivided into Oxfordian and Kimmeridgian - this subdivision, however, was based exclusively on lithological criteria derived from well cuttings and well logs interpretation, without any biostratigraphical support (Dudek and Wójcik, 2011; Dudek et al., 2011; Lach, 2011a, 2011b; Szwed and Wójcik, 2011a, 2011b). As a result, formation tops from new and legacy wells are not stratigraphic equivalents.

Because of those ambiguities exact stratigraphic position of the Upper Jurassic carbonate buildups analysed in this paper remains unclear. Results of recent biostratigraphic studies from the nearby area indicate that the age of similar carbonate buildups ranges from Oxfordian up to Kimmeridgian, and sometimes even up to lower Tithonian (Matyja and Barski, 2007; Matyja, 2009), and it might be assumed that a similar stratigraphic changes might be needed in the Pińczów area described in this paper. It should be stressed however that the precise stratigraphic position of the studied Upper Jurassic carbonate

succession does not have any impact on interpretation of the seismic data presented in this paper; revised stratigraphic schemes might in the future allocate seismically identified carbonate buildups into slightly different Upper Jurassic stratigraphic units.

## 3.2 Seismic data

Two types of seismic data were used in this study: (a) longer legacy profiles, acquired in the early 1990's, located in the central part of the Miechów Trough, (b) short new profiles acquired by SLE in 2011 (Fig. 5). Seismic data was stacked and time migrated, although some seismic artefacts such as diffraction "smiles" are still visible. Beneath the massive carbonates, a velocity pull-up effect could be observed that is distorting the geometry of the pre-Jurassic basement.

Seismic vertical resolution for the Upper Jurassic interval is 10–20 m for the SLE profiles, and 20–30 m for the older legacy lines.

## 3.3 Methodology of well and seismic data integration and interpretation

Precise well-to-seismic tie was based on synthetic seismograms calculated using sonic and density logs for key calibration wells Chopin-1, Belvedere-1 and Michałów-3. Well-to-seismic tie using synthetic seismograms was also carried out for supporting wells (Węchadłów-1, Lipówka-1); however, due to lower quality of the sonic logs, the accuracy of this correlation was significantly lower. Synthetic seismograms allowed correlation of depth well log data (stratigraphy, lithology) with time (TWT) seismic data.

The first phase of seismic data interpretation was carried out for all the seismic profiles. It included identification of main stratigraphic horizons (top Paleozoic, top Triassic, top Middle Jurassic, top Upper Jurassic, top Cenomanian), and the main faults.

The second phase of seismic data interpretation was focused on the Upper Jurassic interval, details of its depositional architecture, and local fault pattern, and included interpretation of all key seismic horizons within the Upper Jurassic succession, analysis of reflection patterns, and recognition of seismic facies related to organic buildups and the surrounding deposits. The seismo-stratigraphic interpretation was carried out for the short SLE lines, and partly for the legacy lines, in the close vicinity of the buildup complexes.

## 4 Results

### 4.1 Well to seismic tie

Synthetic seismograms were calculated using statistical wavelets, with a dominant frequency of 30-35 Hz; wavelet length varied between 120-150 ms (Figs 6 and 7). The two deepest wells (Lipówka-1, Węchadłów-1) that provided information on the top of the Paleozoic basement, top Triassic and top Middle Jurassic were calibrated by the Michałów-3, Lipówka-1 and Węchadłów-1 wells. Other stratigraphic boundaries – top of Upper Jurassic (J3), and top of Cenomanian (Kcn) – were tied to seismic data using data from all five wells used in this study.

## 4.2 1D seismic stratigraphic analysis

For the Chopin-1 and Belverede-1 wells, a detailed 1D seismic stratigraphic analysis was carried out in order to distinguish
the main seismo-stratigraphic units within the Upper Jurassic interval, and to define relationship between the seismic data and lithology and facies of the Upper Jurassic succession. The seismic stratigraphic 1D analysis was conducted using synthetic seismograms calculated for wells Chopin-1 (Fig. 6) and Belvedere-1 (Fig. 7). The precise time-depth model derived from the synthetic seismograms allowed for a detailed correlation of formation tops, well-log data and lithological profile with seismic data. The analysed well log data included gamma-ray, sonic, density and impedance curves. Lithological profiles for both
wells were constructed using well-log data and information from core cuttings (Dudek and Wójcik, 2011; Dudek et al., 2011; Lach, 2011a, 2011b; Szwed and Wójcik, 2011a, 2011b).

As a result, the top of the massive limestones, associated with the carbonate buildups, was defined on seismic data. Also, a correlation of well data with the main depositional systems within the study area following Gutowski et al. (2005) was completed.

Results of 1D seismic-stratigraphic analysis for the Chopin-1 well are shown in Figure 6. Formation tops for the Chopin-1 well included the top of the Cenomanian (Kcn), drilled at 587 m, and top of the Upper Jurassic (J3) drilled at 775 m. Top of the Upper Jurassic is an erosional surface above which the Upper Cretaceous (Cenomanian and younger) rocks were deposited.

Lithological profile for the well Chopin-1 shows that the topmost part (775–843 m) of the Upper Jurassic succession is rather diverse, comprising limestones, marly limestones, marls and claystones (Fig. 6). This lithological diversity is reflected
in variable seismic image - Upper Jurassic top is related to high-amplitude positive seismic horizon generated due to a pronounced lithological contrast between the Cenomanian sandstones and the Upper Jurassic limestones. Beneath the Cenomanian, the Chopin-1 well encountered about 41 m of the Upper Jurassic limestones, mostly white to light grey and medium to hard. This interval could be interpreted as a mainly oolitic and platy limestone dominated succession, well known from the Miechów Trough (cf. Złonkiewicz, 2009). Below, a succession of calcareous claystone, marl and marly limestones
of a total thickness of about 27 m is present. Claystones, marls and marly limestones are expressed by high readings on the gamma-ray log due to the increased content of clay minerals, so this interval (the marly zone in Fig. 6) can at least partly be correlated with the marly facies, including the Lowermost Marly Horizon, of Kutek (1968, 1994). Two lithological intervals described above are characterised by generally high amplitudes of the seismic wavefield (Fig. 6), due to a strong vertical velocity contrasts between the uppermost limestone package and the marly zone below and frequent alterations of marls and
marly limestones. Within the topmost part of the Upper Jurassic succession, the seismo-stratigraphic unit termed J3U was distinguished (Fig. 6). It is characterised by high amplitude seismic horizons. It corresponds mainly to the oolitic-platy limestone succession (Fig. 6). Within this unit, four seismic horizons have been interpreted: 1J3U, 2J3U, 3J3U, 4J3U. The horizon 1J3U corresponds to the very high amplitude negative reflection that is possibly interfered with the above-lying Upper Jurassic top horizon. Its amplitude might be also increased by vertical lithological changes (marl-limestone alternations?)
within the oolitic limestone interval, which is marked by a single peak on the gamma-ray log. The 2J3U horizon represents a

very high amplitude positive reflection which can be associated with a significant increase of seismic velocities (from about 4500 to 5500 m/s), related to those lithological diversity of the oolitic-platy succession. The 3J3U horizon exhibits high amplitude negative reflection which corresponds to a sharp lithological contrast between the oolitic limestones and the marl-claystone formation associated with the upper part of the marly zone. The 4J3U horizon is expressed by a strong positive reflection related to vertical lithological variations within the lower part of the marly zone (from marls to marly limestones). The interval located between the 3J3U and 4J3U horizons is characterised by high values on the gamma-ray log, which indicate a marly zone. However, because of seismic tuning effects, probably caused by frequent marl-limestone alternations, a more precise identification of the marly zone is difficult.

Below the marly zone, a thick (approximately 150 m) succession of hard limestones was drilled (Fig. 6). This succession is related to the massive limestones that commonly form carbonate buildups (see e.g. Matyszkiewicz, 1993; Matyja and Wierzbowski, 2006). The top of the buildup (tcb) on the synthetic seismogram and seismic data is related to relatively low-amplitude positive reflector, probably due to destructive interference from shallower enveloping boundaries (J3U unit). The massive limestone succession is seismically rather homogeneous (Fig. 6).

In Belvedere-1, the entire Upper Jurassic section located below the top of the carbonate buildup (tcb) is more heterogeneous than in the Chopin-1 well (Fig. 7). The top of the carbonate buildup was located at the top of the massive limestone succession. According to the drilling report (Dudek and Wójcik, 2011; Lach, 2011b), the massive limestone succession could be subdivided into two parts by a package of moderately hard platy-like limestones encountered at about 915–935 m. Similarly to the Chopin-1 well, above the carbonate buildup complex the marly zone is present in the Belvedere-1 well, comprising mainly marls and marly limestones about 25–35 metres thick. The marly zone is lithologically diversified which is clearly illustrated by the gamma-ray log as well as the sonic log (Fig. 7). Above the marly zone, a section comprises diverse oolitic-platy limestone deposits (about 50 m thick), which belongs to the uppermost part of the Upper Jurassic, and this interval is associated with the interpreted seismo-stratigraphic J3U unit (Fig. 7). Seismic horizons for both the marly zone and the J3U interval are influenced by intensive intra-bedded signal interferences. This is possibly related to the presence of marl-limestone alternations.

## 4.3 Interpretation of seismic data

All the key seismic horizons (top Paleozoic, top Triassic, top Middle Jurassic, top Jurassic, top Cenomanian) have been interpreted on legacy and new (SLE) seismic profiles. However, due to lower seismic resolution of the legacy data, intra-Upper Jurassic seismic horizons associated with the J3U unit have been interpreted only using new SLE profiles.

The present-day structure of the study area is dominated by reverse faulting along the fault zones deeply rooted in the Paleozoic and older basement (Figs 8–10). Some of these faults might have been active in the Late Jurassic, but clearly their main phase of activity was associated with the Late Cretaceous–Paleogene regional inversion of the Polish Basin (cf. Scheck-Wenderoth et al., 2008; Krzywiec et al., 2009).

Pre-Mesozoic (Precambrian to Carboniferous) rock complexes belong to the Małopolska Block (Żelaźniewicz et al., 2011). It is covered by the Triassic and Middle Jurassic deposits formed within the marginal part of the Polish Basin. The Upper

Jurassic succession exhibits considerable lateral thickness changes caused by variable Late Jurassic local subsidence patterns
(cf. Złonkiewicz, 2006), and later erosion. It gradually thickens towards the north-east, i.e. towards the Holy Cross Mts., where the axial, most subsiding part of the Polish Basin, the Mid-Polish Trough, was located (Figs 8–10). The Jurassic–Cretaceous boundary is related to a subtle angular unconformity or disconformity that truncates the Upper Jurassic strata. The Lower Cretaceous is not present in the study area. The fault-related folding related to inversion of this segment of the Polish Basin could be observed for the entire Upper Cretaceous (Cenomanian–Maastrichtian) succession and indicates latest Cretaceous–
Paleogene age of inversion.

The Upper Jurassic isolated carbonate buildups have been originally identified using legacy seismic profiles. Carbonate buildup drilled by the Chopin-1 well is characterised by the most significant positive relief (Fig. 8). Another organic buildup is located approximately 2 km towards the north-east (Fig. 8). Both these buildups are characterised by a mound-shaped reflection pattern, a drape effect above the structure, and characteristic "depositional wings" associated with the buildup's
edges. Another profile illustrates relatively smaller but a strongly mound-shaped carbonate buildup (Fig. 9). A significant drape effect reaching up to the Cenomanian deposits could be observed above this buildup. Depositional wings are also clearly visible. Identification of the base of this structure is ambiguous. Other examples of the Upper Jurassic carbonate buildups identified on legacy data are shown in Fig. 10. One of those relatively small buildups is located in close vicinity to the Michałów-3 well.

Lateral seismic facies changes within the Upper Jurassic succession are clearly visible on all analysed seismic profiles. Mound-shaped seismic facies that represent carbonate buildups laterally pass into the parallel and continuous seismic reflections related to bedded carbonate deposits.

Much more detailed information on Upper Jurassic carbonate buildups was provided by new SLE seismic data (Figs 11–15). Figure 11 shows results of the detailed interpretation of one of these seismic profiles that was acquired directly above the
large Upper Jurassic carbonate buildup that was drilled by the Chopin-1 well. The carbonate buildup is characterised by chaotic, low-amplitude seismic reflections. Estimated lateral extent of this buildup is up to 1 km. The thickness of the J3U interval is different on both sides of the buildup – it increases from its east side, where several onlapping horizons are visible. This might be related to local syn-depositional faulting within deeper substratum. The base of this buildup is not clearly imaged due to (1) strong wavelet interference, (2) reflections from the buildup's edges, (3) limited seismic resolution.

Several small-scale faults have been interpreted on this seismic profile. Deeper faults that dissect Paleozoic–Triassic–Middle Jurassic interval might be partly related to older phases of tectonic evolution of the area. However, it should be stressed that time seismic data might also suffer from local velocity effects such as velocity pull-up beneath the massive – i.e. seismically fast – carbonates. Therefore, the interpreted geometry beneath the carbonate buildups should be treated with certain caution and not regarded as an exact representation of the sub-Upper Jurassic structure.

Upper Jurassic carbonate buildup drilled by the Belvedere-1 well is shown in Fig. 12. This structure does not exhibit such strong positive relief as the buildup shown in Fig. 11, and its outline is less visible. This carbonate buildup consists of two massive limestone successions, separated by platy-like limestone strata (Fig. 7), and this might be one of the reasons for less

clear seismic imaging. The western edge of this carbonate buildup is dissected by a normal fault, across which a slight thickness increase of the J3U unit is observed, suggesting syn-depositional activity.

The highest amplitudes and the most continuous seismic horizons are observed for the J3U unit (Figs 11 and 12). This might be related to a sharp lithological contrasts within this interval caused by the occurrence of limestones interbedded by marls and marly limestones of the marly zone (cf. Figs 6 and 7). Sub-horizontal seismic horizons, associated with bedded carbonates surrounding the carbonate buildups, are also clearly visible (Figs 11 and 12).

    Finally, important differential compaction and related compaction sag effect could be observed above all the identified
carbonate buildups. Carbonate buildups, generally represented by rigid, massive limestones, are more resistant to compaction, while the surrounding bedded carbonate facies are much more prone to compaction. This effect can be very clearly seen on the seismic profile shown in Fig. 13. It is expressed by: (1) the drape effect above the buildup - an evidence of lower compaction (typical for resistant carbonate buildup deposits), and (2) compaction sag as evidence of higher compaction, typical for bedded carbonates surrounding the buildups. This seismic pattern could be observed for the Upper Jurassic succession and, although
to a lesser degree, also within the lowermost part of the Upper Cretaceous succession. Differential compaction may have also led to formation of some of the normal faults along the borders of the carbonate buildups (Figs 11 and 13).

    The velocity pull-up effect observed beneath the carbonate buildups (cf. Fig. 13) results from lateral seismic velocity contracts between the massive and stratified (bedded) carbonates. The interval velocity of the massive limestones, drilled by modern Chopin-1 and Belvedere-1 wells, is about 5000–5000 m/s and is significantly higher in comparison to seismic velocity
obtained from the Michałów-3 or Lipówka-1 legacy wells for the corresponding stratified deposits that are in order of ca. 3800-5000 m/s. However, it should be taken into account that velocity information from these old wells should be treated only tentatively, due to their uncertainty resulting from the lower quality of older well-logging data. Expected lateral seismic velocity variations between the massive and bedded carbonates often exceed 10% and might be responsible for producing some velocity pull-ups beneath the seismically faster carbonate buildups. Then, it is probable that at least for some of the
morphological heights situated beneath the carbonate buildups in the analysed time seismic data, velocity pull-ups might have distorted their true geometries. The similar role of high-velocity reefal intervals in production of velocity pull-up effects beneath the carbonate buildups was described for time seismic data characterising the large Miocene buildups in Luconia, Malaysia (e.g. Zampetti et al., 2004; Rankey et al. 2019) or numerous isolated buildups from the north-west shelf of Australia (Saquab and Bourget, 2016).

The lateral extent of the carbonate buildups identified using seismic data from the central part of the Miechów Trough is in the range of 400–1000 m, the present day total height of most structures is around 150–200 m. Yet, present day observed cumulative height of the two largest complexes, drilled by the Chopin-1 and Belvedere-1 wells, probably exceeds 250 m. However, identification of the base of the buildups was ambiguous due to rather poor seismic imaging of the lowermost part of the large buildup complexes (Figs 11 and 12). Both structures are hundreds of metres long (even up to 1 km, Figs 11–13).

## 5 Discussion

### 5.1 Carbonate buildups on seismic data – regional context

Results presented in this paper illustrate how and to what extent seismic data can be used for analysis of carbonate depositional systems, in particular for identification of the carbonate buildups and the enveloping strata. In this study, several large Upper Jurassic carbonate buildups in the Miechów Trough (southern Poland) have been seismically identified and characterised. Possible occurrences of carbonate buildups in the study area has been already tentatively proposed by several authors (cf. Gutowski et al., 2005; Matyja, 2009; Złonkiewicz, 2009). However, so far no direct evidence of their presence in this part of the basin has been presented. In southern Poland, where the Upper Jurassic strata is covered by thick Cretaceous and younger deposits, such as in the Miechów Trough, previous studies of these deposits were carried out using information from older research wells only (cf. Złonkiewicz, 2006, 2009). Availability of deep wells in this part of the basin, including the study area, is however, insufficient for detailed analysis of the geometry and architecture of the carbonate depositional system, in particular for identification of the carbonate buildups. In comparison to adjacent areas in Poland and central and western Europe, where the Upper Jurassic is well-known from outcrops (cf. Leinfelder et al., 1996; Matyszkiewicz, 1997a), a carbonate succession in the Miechów Trough remained until recently much less recognised. Seismic data described in this paper allowed for identification of large carbonate buildups and surrounding enveloping strata, and therefore provided new crucial information on the Late Jurassic depositional system in this part of the basin. Results of this study could also be used as a more universal reference point for seismic studies of carbonate depositional systems of different ages from different sedimentary basins.

The seismically interpreted carbonate buildups from the study area formed part of the vast Late Jurassic carbonate depositional system that developed along the northern, passive shelf of the Tethys, forming in Europe a belt extending from Portugal through Spain, France, southern Germany, Poland to Ukraine and Romania (Leinfelder et al., 1996). Quality of the presented seismic examples is quite unique in comparison to other few papers dealing with seismic interpretation of Upper Jurassic carbonate buildups in central and western Europe. In comparison to the Upper Jurassic carbonate buildups seismically recognised from Southern Germany (cf. Hartmann et al., 2012; Lüschen et al., 2014; see Fig. 2) buildups described in this paper are much better imaged on seismic data. The present day observed cumulative heights of carbonate buildups from the Miechów Trough are distinctly larger than in the seismically described reefs from the Bavarian Molasse Basin, for which a total thickness of the reef succession does not exceed 180 m (Hartmann et al., 2012). This confirms that carbonate sedimentation in the Polish part of the northern Tethyan shelf was more intense than in southern Germany (cf. Matyja and Wierzbowski, 1996). Observed vertical size of the carbonate buildups described in this study is similar to the Upper Jurassic reefs recognised on seismic data from the Western Caucasus and Black Sea region (Afanasenkov et al., 2007; Guo et al., 2011). This suggests that in both areas local depositional environment (including palaeo-bathymetry and subsidence) was at least generally similar.

The Upper Jurassic carbonate buildups have been also recognized using seismic data in Poland, 40–60 km south from the study area within the southern segment of the Miechów Trough that is covered by Miocene sediments of the Carpathian

Foredeep Basin (Misiarz, 2003; Gliniak and Urbaniec, 2005; Gliniak et al., 2005; Jędrzejowska-Tyczkowska et al., 2006).
Vertical and lateral size of those structures is generally comparable to size of carbonate buildups described in this paper. This
suggests that growth of all these structures took place in a relatively unified depositional environment that characterized this
part of the basin. Recently, Urbaniec (2019) provided seismic examples of Upper Jurassic carbonate buildups of similar size
that are located about 50 km south-east from the study area. Those buildups are characterised by complex geometries and
probably consist of several levels of the massive limestones.

Presented results of seismic interpretation of carbonate buildups have also more universal application. They can be used as
a reference point for analysis of carbonate buildups and elements of depositional system using seismic data from other
sedimentary basins. The quality of the seismic image is comparable to some case studies of this type from various areas in the
world (cf. Elvebakk et al., 2002; Zampetti et al., 2004).

## 5.2 Geometry and depositional architecture of the Upper Jurassic basin

Depositional architecture of the Upper Jurassic carbonate succession in the study area recognised on seismic data (Figs 11–
15) resembles a classic carbonate system well-known from outcrops located within the adjacent Kraków-Częstochowa Upland
(Figs 3 and 4A). It is characterised by presence of carbonate buildup complexes surrounded by diverse bedded carbonate facies
(Dżułyński, 1952; Matyja and Wierzbowski, 1996, 2006; Matyszkiewicz et al., 2012; Krajewski et al., 2018). The Upper
Jurassic succession in the Kraków-Częstochowa Upland (Fig. 3 and Fig. 4a) is characterised by strong local vertical and lateral
thickness, and facies variability. This is mainly related to differentiated relief at the top of Paleozoic substratum associated
with local differentiation in subsidence caused by the occurrence of Permian intrusions, syn-sedimentary tectonics, and local,
mostly aggradational growth of organic buildups, as well as differential compaction of carbonate sediments (Matyszkiewicz,
1999; Matyszkiewicz et al., 2006, 2012, 2016; Kochman & Matyszkiewicz, 2013; Matyszkiewicz and Kochman, 2016). The
Upper Jurassic succession in the Kraków-Częstochowa Upland consists of: (1) bedded facies, (2) massive facies, and (3)
deposits of gravity flows (Matyszkiewicz et al., 2012). Massive and bedded limestone facies belong to the sponge megafacies
deposits (Matyja and Wierzbowski, 2006). This succession, characterised by the abundant presence of siliceous sponges and
microbial structures, is common within the northern Tethyan shelf margin of central and western Europe in the Late Jurassic
(e.g. Matyja and Pisera, 1991; Matyja and Wierzbowski, 1995, 2006; Wierzbowski et al., 2016; see Fig. 2). In the Kraków-
Częstochowa Upland, massive limestones constitute large carbonate buildup complexes that are surrounded by bedded
limestones and marls that were formed within intra-buildup sub-basins, or much wider (up to several km long) inter-buildup
basins (cf. Matyja and Wierzbowski, 1996). Massive facies (carbonate buildups) passes laterally into bedded facies (Gutowski
et al., 2005, 2006; Matyja and Wierzbowski, 2006, see Fig. 4a). Similar elements of depositional architecture can be observed
on seismic data from the study area (Figs 14–15). Mound-shaped seismic facies that represent carbonate buildups laterally
pass into the parallel and continuous seismic reflections related to bedded carbonate deposits represent intra-buildup sub-basins
(Figs 14–15). Present day cumulative heights (150–250 m) and lateral extents (400–1000 m) of the structures identified on

seismic data from the Miechów Trough are generally comparable with large carbonate buildup complexes known from the Kraków-Częstochowa Upland (cf. Matyja and Wierzbowski, 1996, 2006; Matyszkiewicz et al., 2006, 2012, 2015b).

Presented examples strongly suggest that (1) similar basin geometries (e.g. carbonate buildups, intra-buildup sub-basins), and (2) main facies relationship (i.e. massive facies versus bedded facies) for the Upper Jurassic succession could be reliably distinguished on seismic data from the study area. Seismic image of bedded facies revealed significant vertical lithological

variations which are expressed by high-amplitude continuous seismic reflections (see Figs 14–15). This might be related to strong vertical lithological variability known from equivalent deposits in the Kraków-Częstochowa Upland, where the bedded facies commonly include several meter thick marls and marl-limestone alternations (e.g. Matyja and Wierzbowski, 2006; Matyszkiewicz, 2008). Such distinct lithological contrasts (bedded limestones alternated by marls), are probably responsible for producing these characteristic strong, sub-horizontal seismic horizons (Fig. 16b and d, compare with Figs 14–15).

The results of the seismic interpretation could be correlated with the main Upper Jurassic lithofacies scheme proposed by Złonkiewicz (2009) for the entire Miechów Trough (see Fig. 4b). The uppermost part of the Upper Jurassic interval which corresponds to the J3U seismic-stratigraphic unit, characterised by high-amplitude flat seismic horizons (Figs 6–7 and Figs 14–15) can be related to the shallow-water carbonate platform, represented predominantly by oolitic platy deposits, comprising mainly limestones, marly limestones and marls (Złonkiewicz, 2009; cf. Kutek, 1968; Matyja et al., 1989, 2006; Gutowski et

al., 2005, 2006; Krajewski et al., 2017, see Fig. 4b). Massive limestones, which represent carbonate buildups, might be related to the Massive Limestone Member, and, the bedded facies could refer to the heterogeneous Siedlce Limestone Member (Matyja, 1977; Złonkiewicz, 2009, see Figs 4B and 14). The lower part of the Upper Jurassic interval may also be partly associated with the Morawica Limestone Member and with lowermost marly-dominated strata (Złonkiewicz, 2009; see Fig. 4b).

Above the top of large carbonate buildups (Figs 11–12), the higher gamma-ray log values clearly indicate the presence of marly and marly limestones deposits those are related to the marly zone interpreted from the 1D seismic stratigraphic analysis (see Figs 6–7). This interval might be associated with the disappearance of the organic buildups, and a change in sedimentary conditions related to drowning of the carbonate ramp, which is evident from deposition of deeper-water marly facies (cf. Krajewski *et al.*, 2017; see also Kutek, 1968, 1994).

**6 Conclusions**

This paper provides new important information on the Late Jurassic sedimentation in the Miechów Trough that was located within the transition zone between epicontinental basin of western and central Europe and the Tethys Ocean. This area has been hitherto less recognised in comparison to adjacent areas in Poland and central and western Europe, where the Upper Jurassic deposits are well-known from outcrops. The results of seismic data interpretation from the study area proved the

presence of the large carbonate buildups in this part of the Upper Jurassic basin in Southern Poland. The identified carbonate buildups exhibit significant positive relief; lateral extent of particular complexes is in range of 400–1000 m, and observed

cumulative height of the most structures is in range of 150–250 m. This study revealed the distinctive depositional architecture of the carbonate Upper Jurassic succession. The mound-shaped seismic facies that represent large carbonate buildups laterally pass into the parallel and continuous seismic reflections related to the well-bedded basinal deposits (limestones and marls) that

were formed within the intra-buildup sub-basins or wider inter-buildup basins. The geometry and architecture of the Upper Jurassic basin in the Miechów Trough resemble the well-recognised open shelf carbonate depositional system from the adjacent Kraków-Częstochowa Upland. Seismic unit J3U that overlies the open shelf deposits, represent younger, shallow-water carbonate platform succession. Between the J3U unit and the top of carbonate buildups, the distinctive marly zone has been interpreted from the 1D seismic-stratigraphic analysis and the gamma-ray logs. This interval is represented by deeper-water

facies that might be related to the carbonate platform drowning and demise of the organic buildups. The results shown in this paper fill the gap in recognition of the Late Jurassic palaeogeography of southern Poland and provide well-documented example of a system of carbonate buildups developed in the south-eastern segment of the transition zone between the European Late Jurassic epicontinental basin and the Tethys Ocean.

**Data availability**

The data presented in this study have been provided by oil company and are not publicly accessible.

**Author contribution**

Łukasz Słonka carried seismic interpretation and prepared the manuscript, Piotr Krzywiec participated in analysis and discussion of the final results and supervised preparation of the manuscript.

**Competing interests**

The authors declare that they have no conflict of interest.

**Acknowledgements**

San Leon Energy and PGNiG S.A. are thanked for granting access to the seismic and well data used in this study. IHS Markit kindly provided Kingdom seismic interpretation software. Łukasz Słonka would like to thank Prof. Andrzej Wierzbowski

(PGI, Warsaw), Prof. Jacek Matyszkiewicz (AGH UST, Kraków), and Dr Marcin Krajewski (AGH UST, Kraków) for their literature recommendations and for constructive comments on the early results of this study, and Prof. Joachim Szulc (deceased), Dr Anna Lewandowska and Dr Wojciech Wróblewski (all Jagiellonian University, Kraków) for their help during the fieldwork. Many thanks are due to Aleksandra Stachowska for her help in collecting materials and helpful comments on

the final version of the paper, and to Dr Ashley Gumsley for correcting English. Comments and suggestions from reviewers
Prof. Jacek Matyszkiewicz, Prof. Tadeusz Peryt and Dr Gabor Tari greatly help to finally refine this paper and are highly appreciated.

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

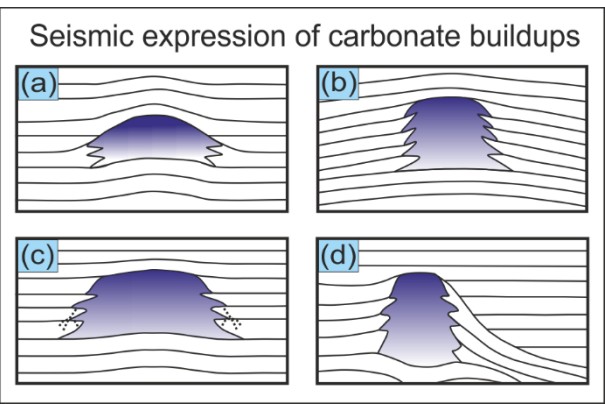

**Figure 1.** Common types of seismic expression of carbonate buildups used for their seismic identification and characteristics (based on Bubb & Hatlelid, 1977; Veeken & Moerkerken, 2013; modified): **(a)** velocity pull-up and differential compaction, **(b)** reflection free with drape effect, **(c)** reflection free with diffractions on edge, **(d)** compaction sag and transgressive onlap.

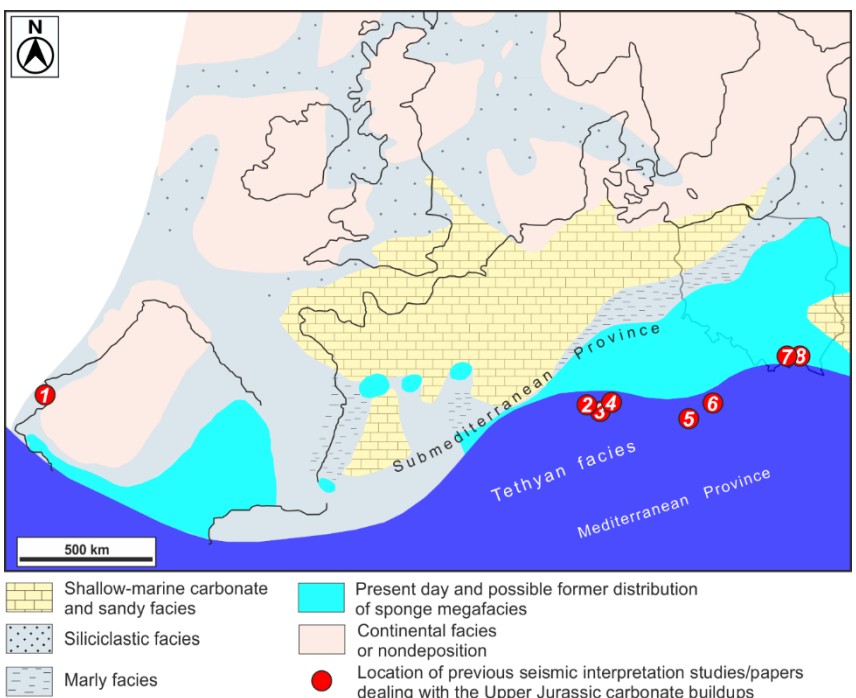

**Figure 2.** Simplified paleogeographic sketch map of central and western Europe for the middle–late Oxfordian (after Wierzbowski et al., 2016); red points show location of the previously published seismic interpretation studies/papers dealing with the Upper Jurassic carbonate buildups from the northern Tethyan shelf margin and adjacent areas (1. Ellis et al., 1990; 2. Bunes et al., 2010; 3. Hartmann et al., 2012; 4. Lüschen et al., 2014; 5. Zimmer and Wessely, 1996; 6. Adámek, 2005; 7. Gliniak and Urbaniec, 2001; 8. Gliniak et al., 2005; see text for more details).

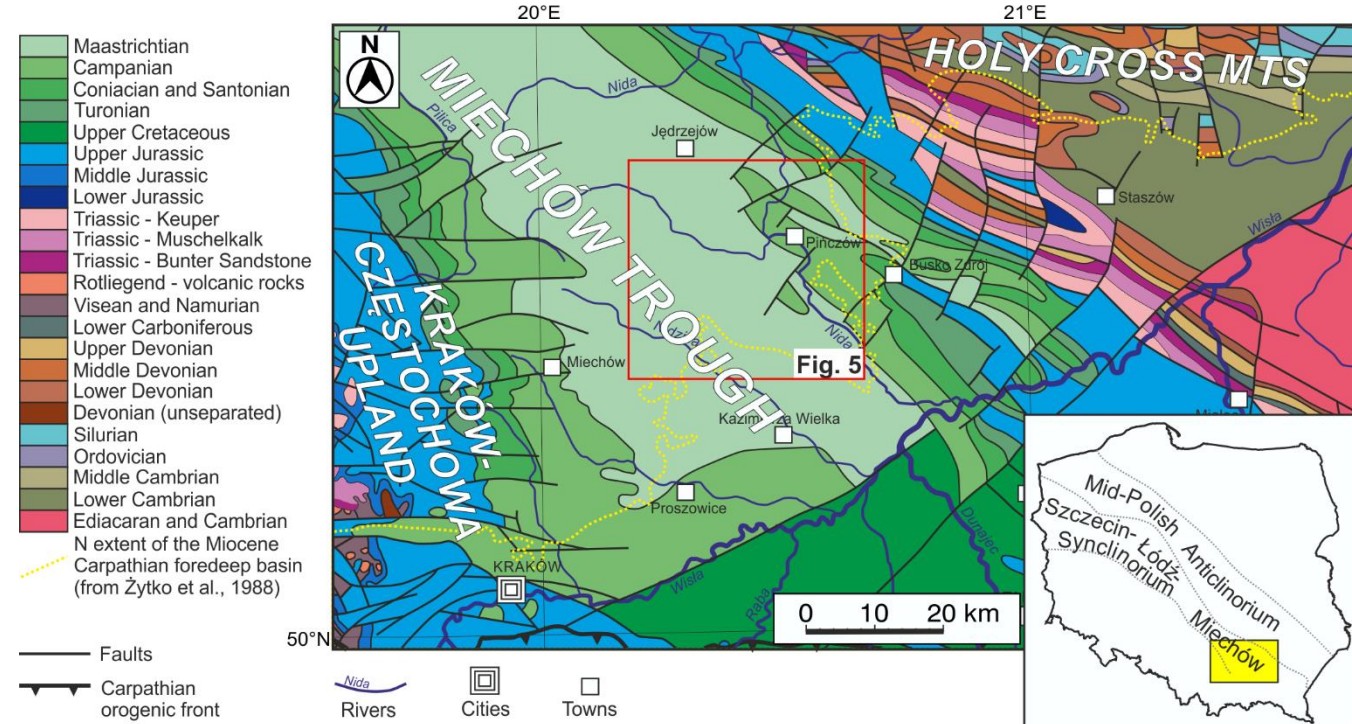

**Figure 3.** Geological map of the Miechów Trough and the adjacent areas (after Dadlez et al., 2000; simplified); the inset map shows location of this unit in Poland (yellow rectangle); the study area is indicated by the red rectangle.

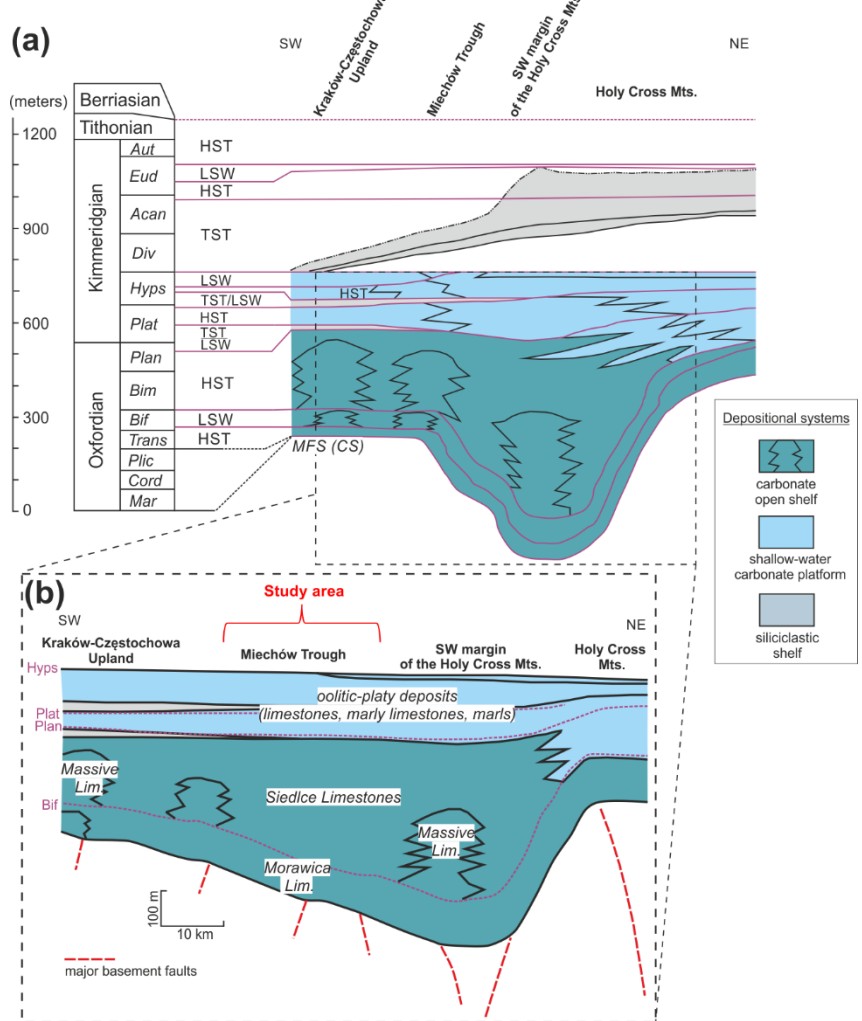

**Figure 4. (a)** Simplified, idealised stratigraphic scheme of the Late Jurassic epicontinental basin in southern Poland including the Miechów Trough, showing main depositional systems and cyclicity (after Gutowski et al., 2005); Submediterranean ammonite zones abbreviations: Mar – Mariae, Cor – Cordatum, Plic – Plicatilis, Trans – Transversarium, Bif – Bifurcatus, Bim – Bimammatum, Plan – Planula, Plat – Platynota, Hyps – Hypselocyclum, Div – Divisium, Acan – Acanthicum, Eud – Eudoxus, Aut – Autissiodorensis); system tracts: HST – highstand, LSW – lowstand wedge, TST – transgressive, MFS (CS) – maximum flooding surface (condensed section); **(b)** details of the Oxfordian–Kimmeridgian interval showing main Upper Jurassic lithological units in the study area and adjacent regions (based on Gutowski et al., 2005, 2006; Złonkiewicz, 2009; simplified).

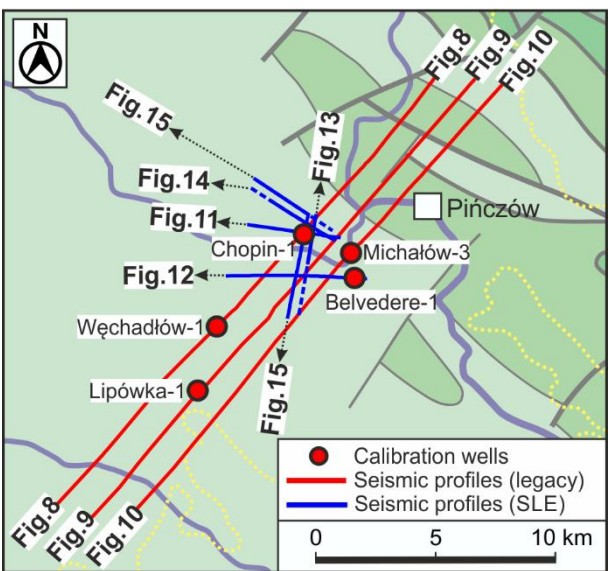

**Figure 5.** Detailed view of the study area with location of wells and seismic data. Solid lines (red and blue) mark the sections of the seismic profiles shown in the Figs 8–15. Background geological map (after Dadlez et al., 2000; Żytko et al., 1988; simplified) from Fig. 3 (see for description).

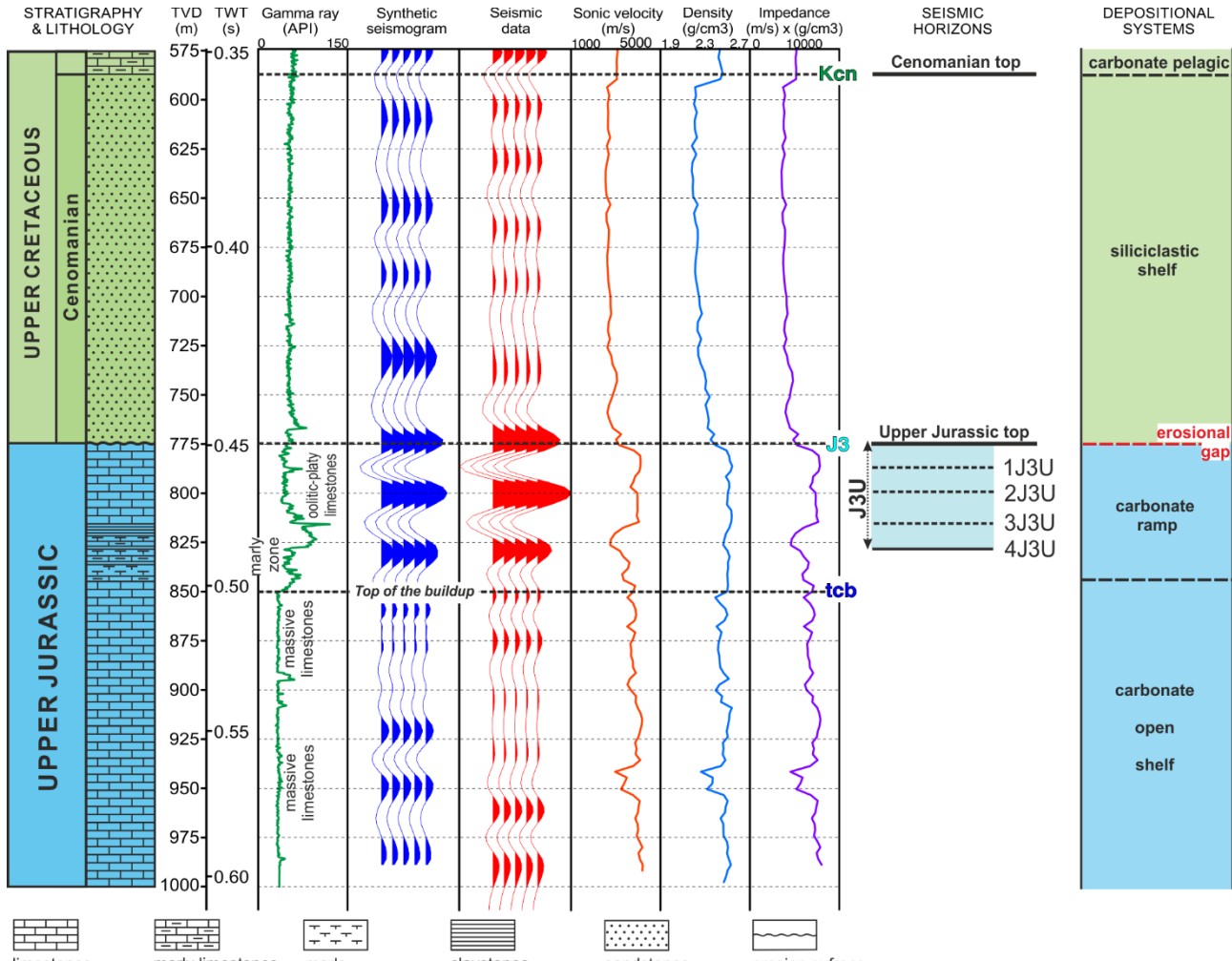

**Figure 6.** Well-to-seismic data correlation together with simplified lithostratigraphic profile for the Upper Jurassic succession and its Cenomanian overburden, Chopin-1 well. 1D seismic-stratigraphic analysis allowed for identification of top of carbonate buildup deposits (represented by massive limestones - tcb), of top Upper Jurassic (J3), of top Cenomanian (Kcn), and of 4 main seismic horizons within the uppermost part of the Upper Jurassic interval located above the buildup deposits (1J3U, 2J3U, 3J3U, 4J3U). Main depositional systems are based on Gutowski et al. (2005) and Krzywiec et al. (2009).

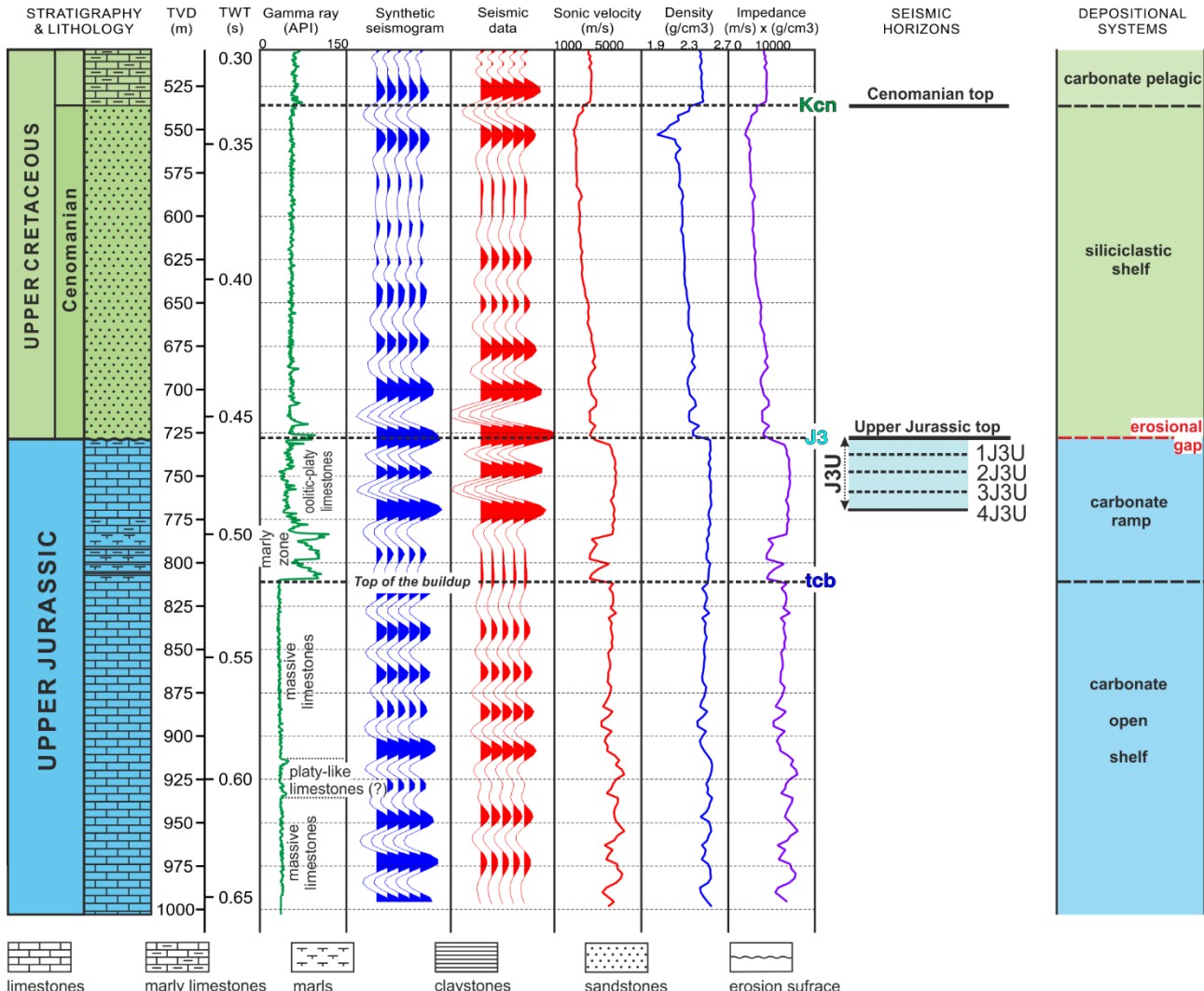

**Figure 7.** Well-to-seismic data correlation together with simplified lithostratigraphic profile for the Upper Jurassic succession and its Cenomanian overburden, Belvedere-1 well. 1D seismic-stratigraphic analysis allowed for identification of top of carbonate buildup deposits (represented by massive limestones - tcb), of top Upper Jurassic (J3), of top Cenomanian (Kcn), and of 4 main seismic horizons within the uppermost part of the Upper Jurassic interval located above the buildup deposits (1J3U, 2J3U, 3J3U, 4J3U). Main depositional systems are based on Gutowski et al. (2005) and Krzywiec et al. (2009).

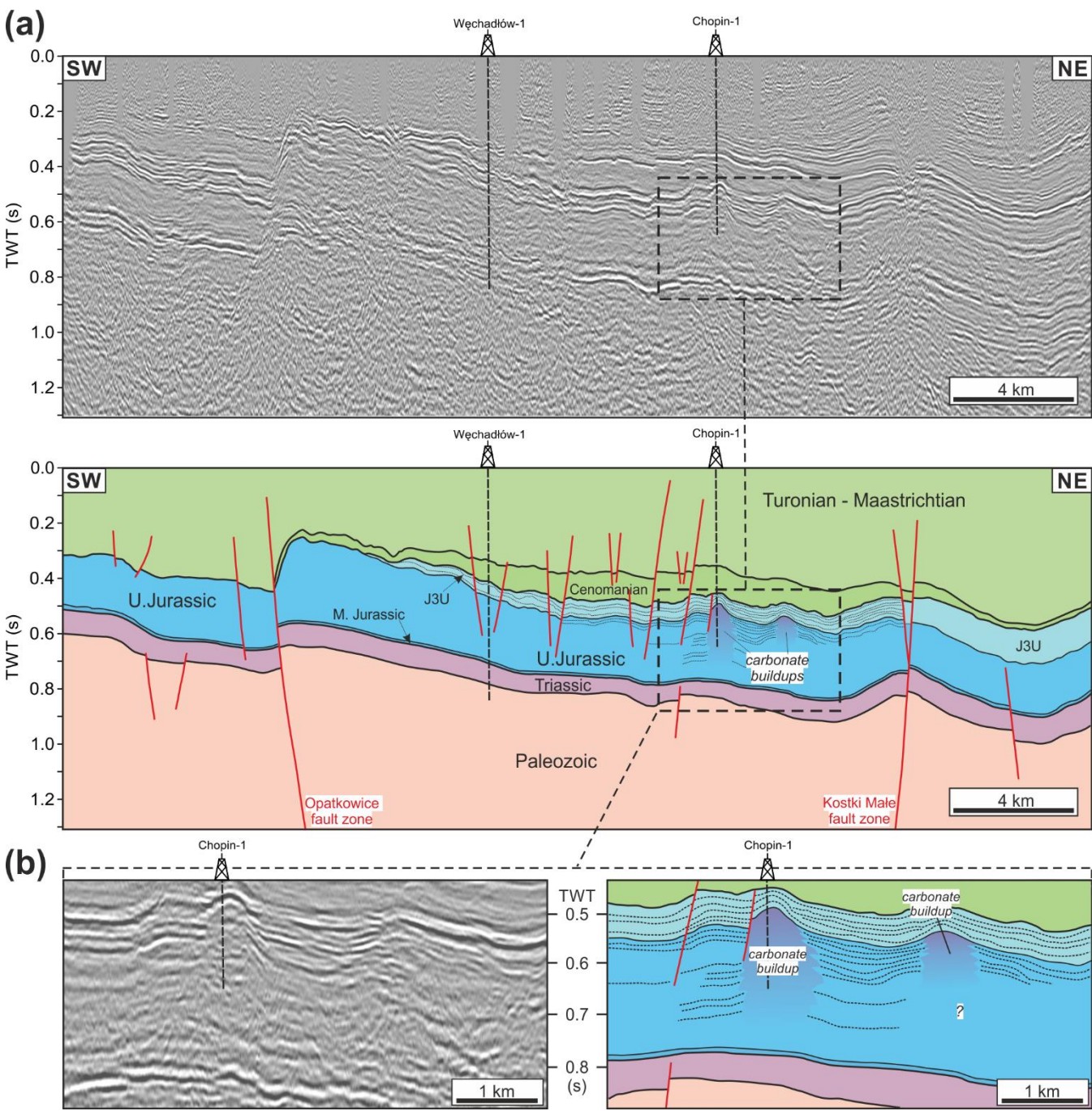

**Figure 8. (a)** Uninterpreted and interpreted seismic profile (12-5-92K) from the Miechów Trough, see Figure 5 for location. Major NW-SE oriented Opatkowice and Kostki Małe fault zones are rooted in the Paleozoic basement and associated with inversion anticlines developed within the Mesozoic cover; **(b)** Two carbonate buildups were identified in this profile; one of them was partly drilled by the SLE Chopin-1 well.

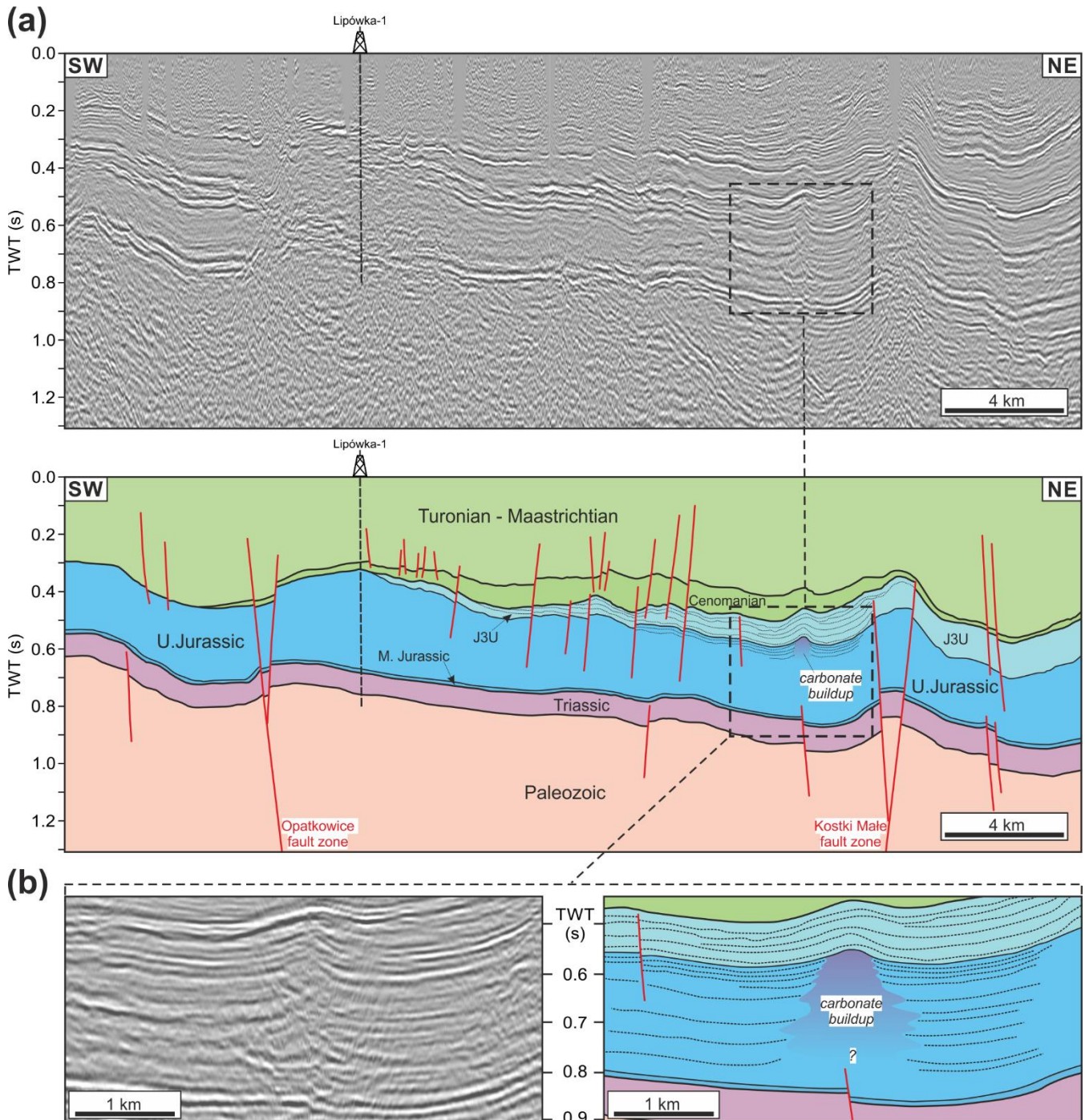

**Figure 9. (a)** Uninterpreted and interpreted seismic profile (11-5-92K) from the Miechów Trough, see Figure 5 for location. Major NW-SE oriented Opatkowice and Kostki Małe fault zones are rooted in the Paleozoic basement and associated with inversion anticlines developed within the Mesozoic cover; **(b)** One isolated carbonate buildup was identified in this profile.

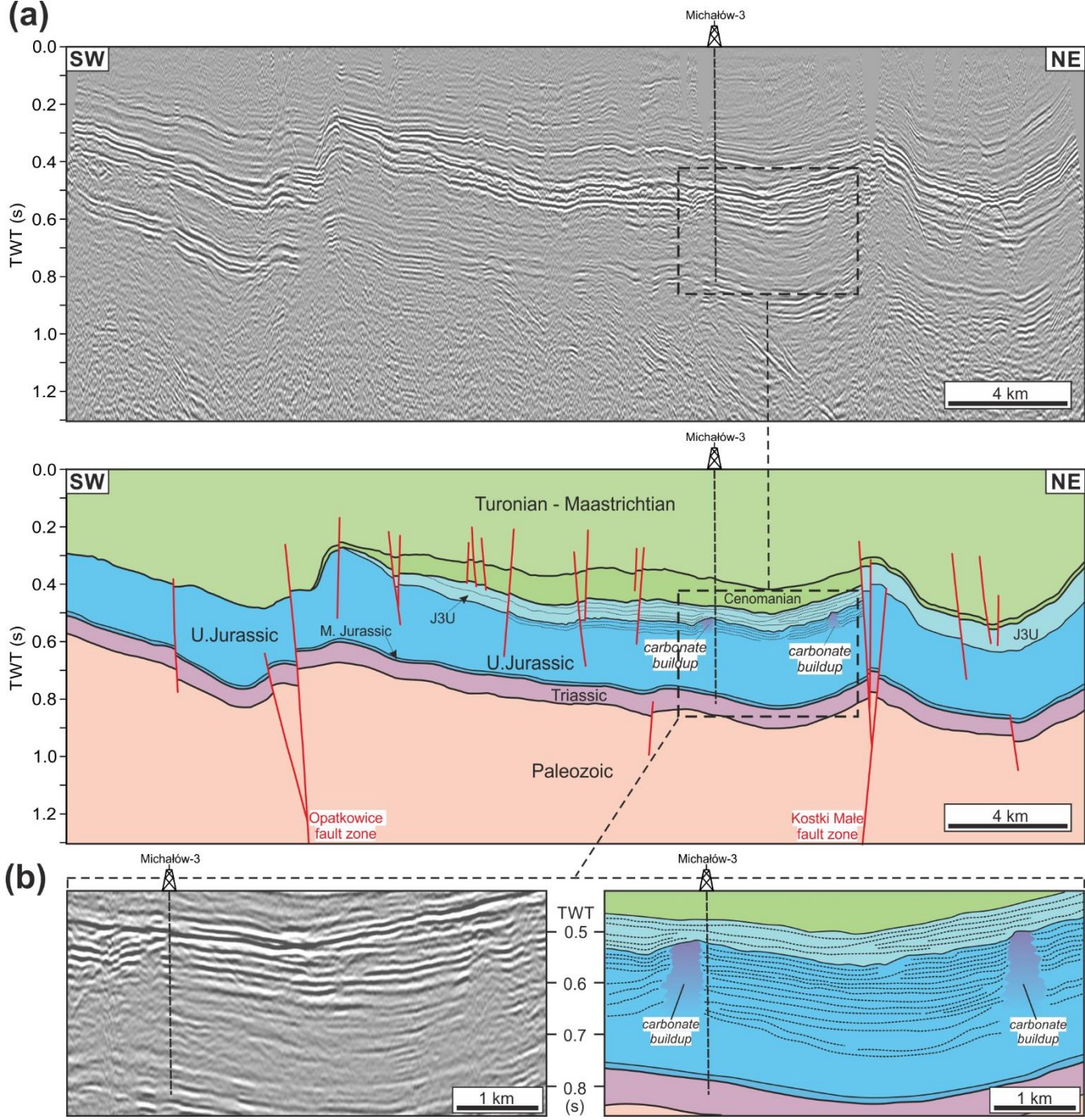

**Figure 10. (a)** Uninterpreted and interpreted seismic profile (10-5-92K) from the Miechów Trough, see Figure 5 for location. Major NW-SE oriented Opatkowice and Kostki Małe fault zones are rooted in the Paleozoic basement and associated with inversion anticlines developed within the Mesozoic cover; **(b)** Two carbonate buildups were identified in this profile.

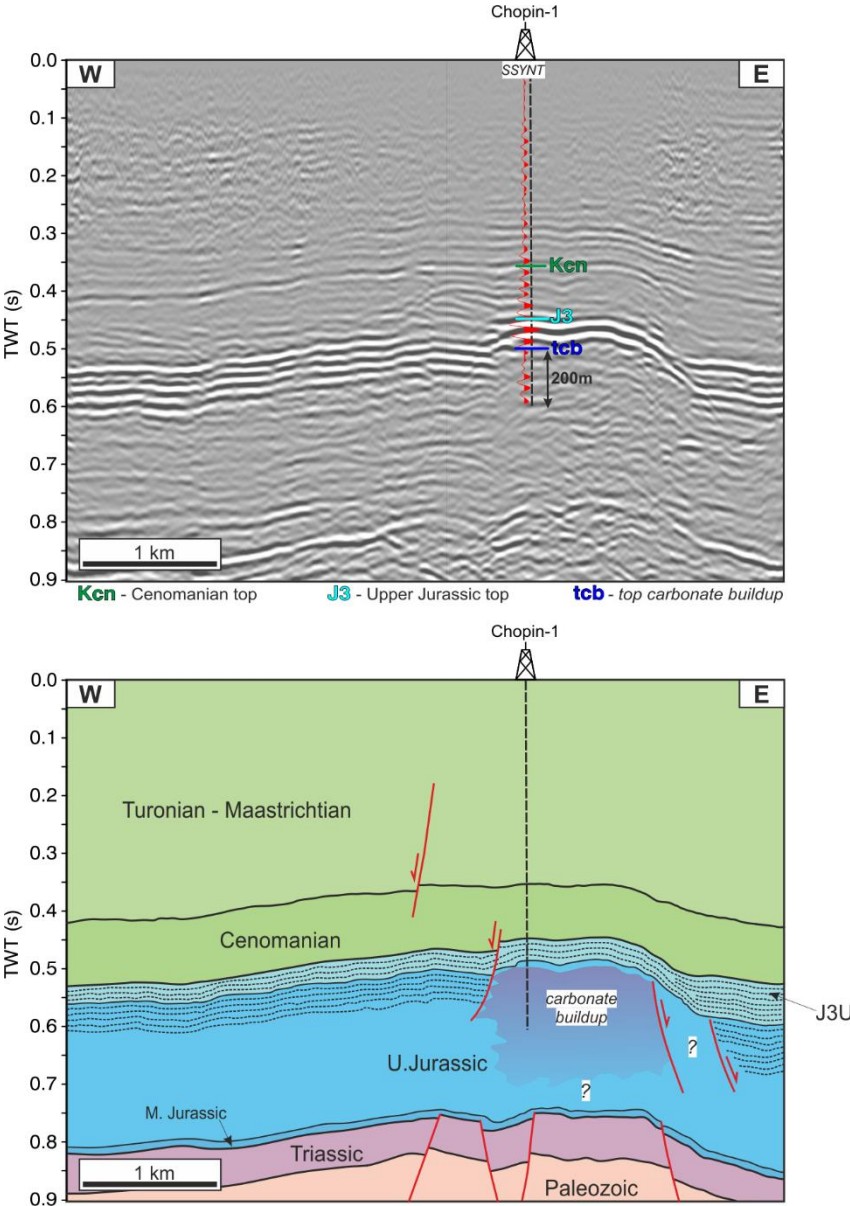

**Figure 11.** Uninterpreted and interpreted seismic profile across the carbonate buildup complex, see Figure 5 for location; this carbonate buildup was partly drilled by the Chopin-1 well. Well tops, interpreted top of carbonate buildup (tcb) and the synthetic seismogram (SSYNT) are shown (see Figure 6 for details).

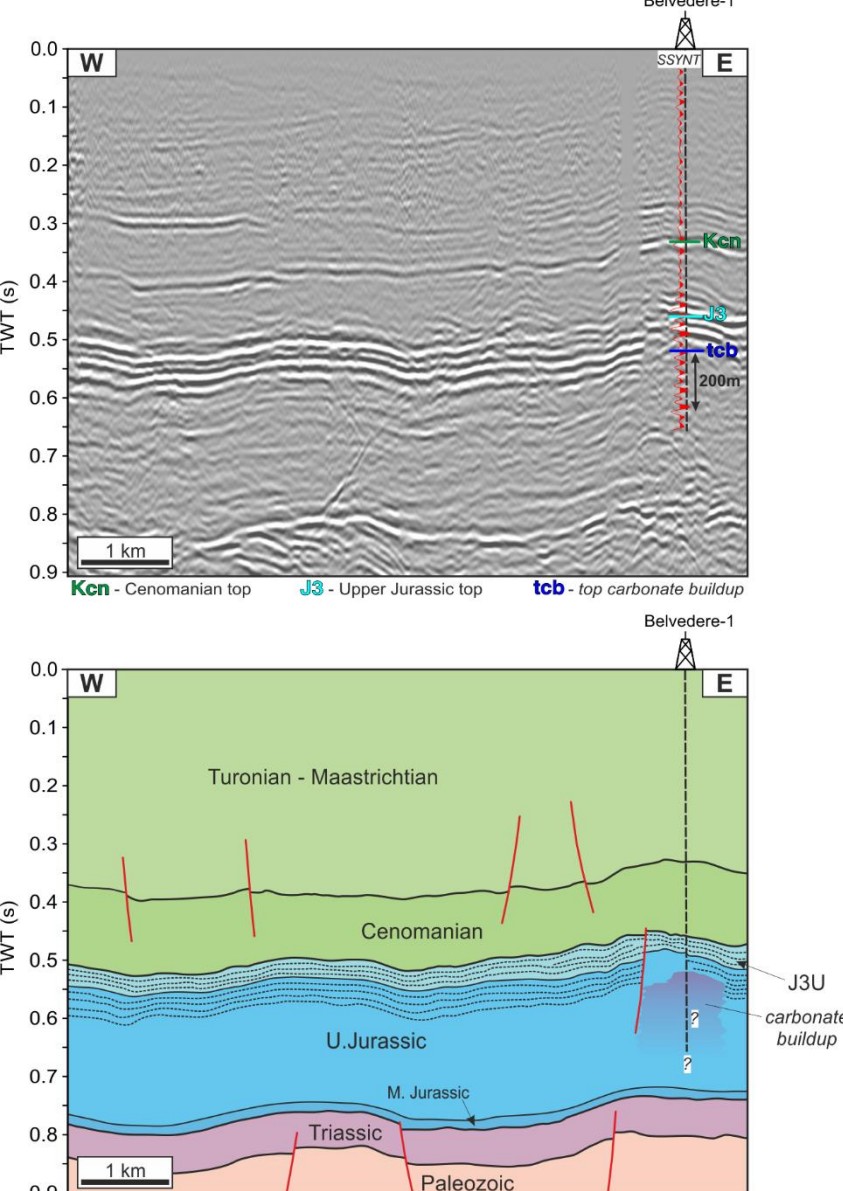

**Figure 12.** Uninterpreted and interpreted seismic profile across the carbonate buildup complex, see Figure 5 for location; this carbonate buildup was partly drilled by the Belvedere-1 well. Well tops, interpreted top of carbonate buildup (tcb) and the synthetic seismogram (SSYNT) are shown (see Figure 7 for details).

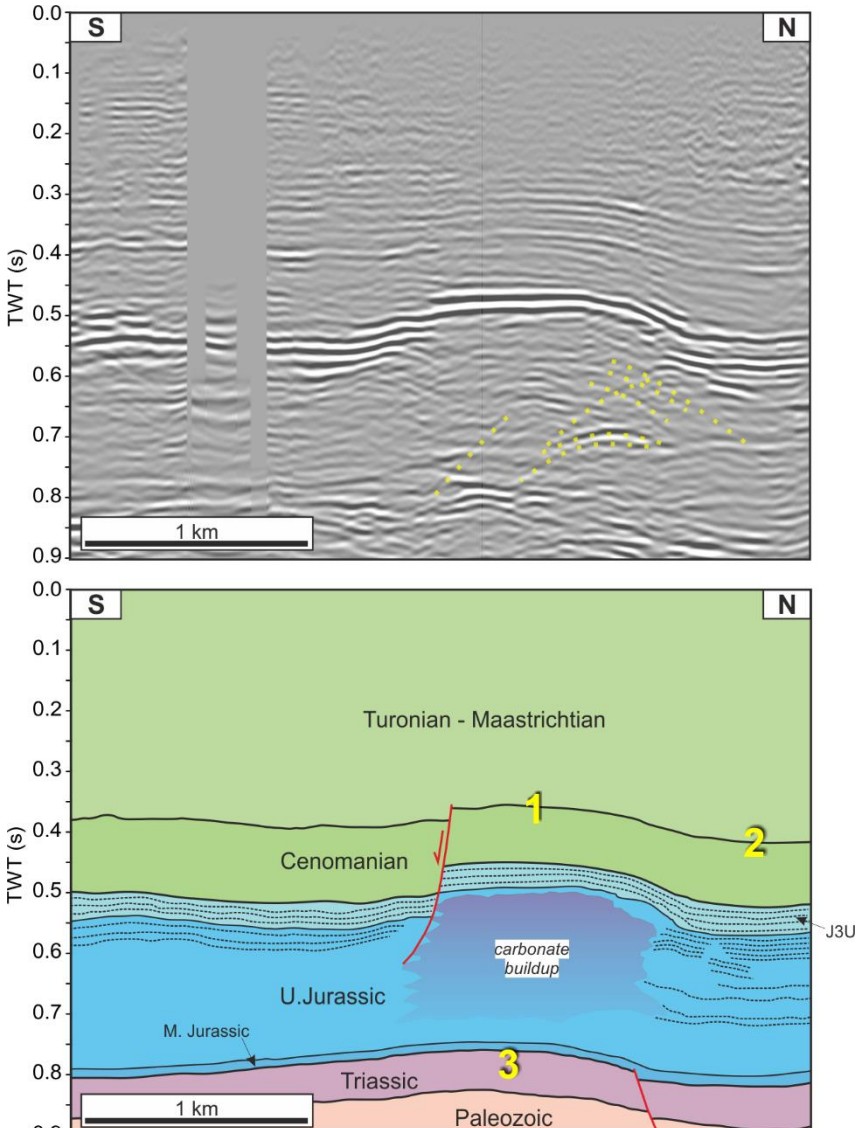

**Figure 13.** Uninterpreted and interpreted part of seismic profile across the carbonate buildup complex, see Figure 5 for location. The effect of differential compaction between the carbonate sediments (generally much higher for bedded carbonate facies, very low for massive limestones) can be clearly seen. Usually, these effects could be also visible within the younger, Upper Cretaceous overburden. As result, the younger strata also partly exhibit drape reflections (1) and compaction sag effect (2). Other characteristic seismic indicators can be also observed, i.e. the velocity pull-up effect for horizons below the buildup's base (3), and the diffraction on buildup's edges (yellow dotted lines marked on the uninterpreted profile). Differential compaction may have also led to formation of normal faults along the borders of carbonate buildup.

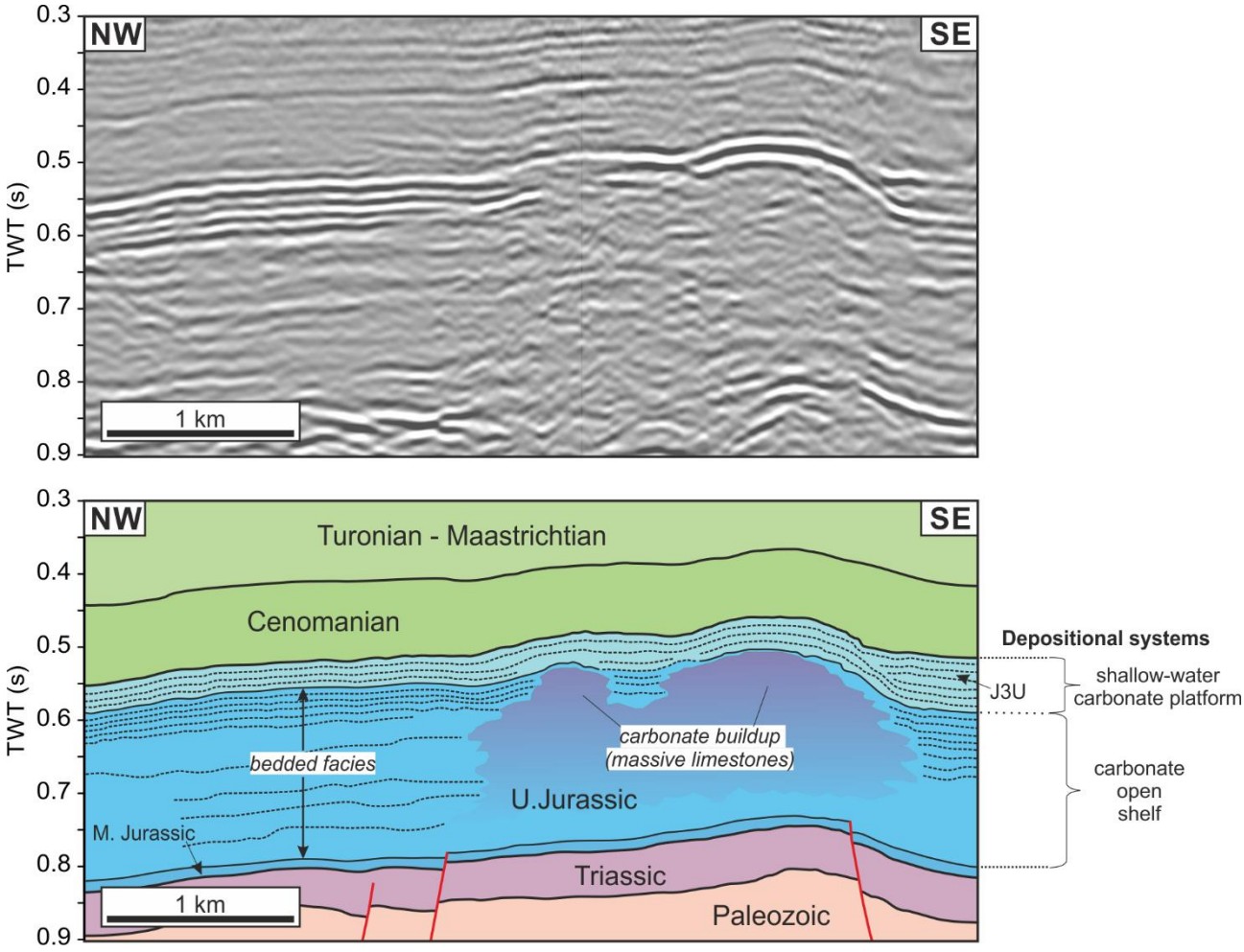

**Figure 14.** Uninterpreted and interpreted part of seismic profile across the carbonate buildup complex, see Figure 5 for location. Lateral seismic facies changes correspond to main facies changes within the Upper Jurassic: bedded facies (represented by diverse bedded limestones and marls) - massive limestones (carbonate buildups).

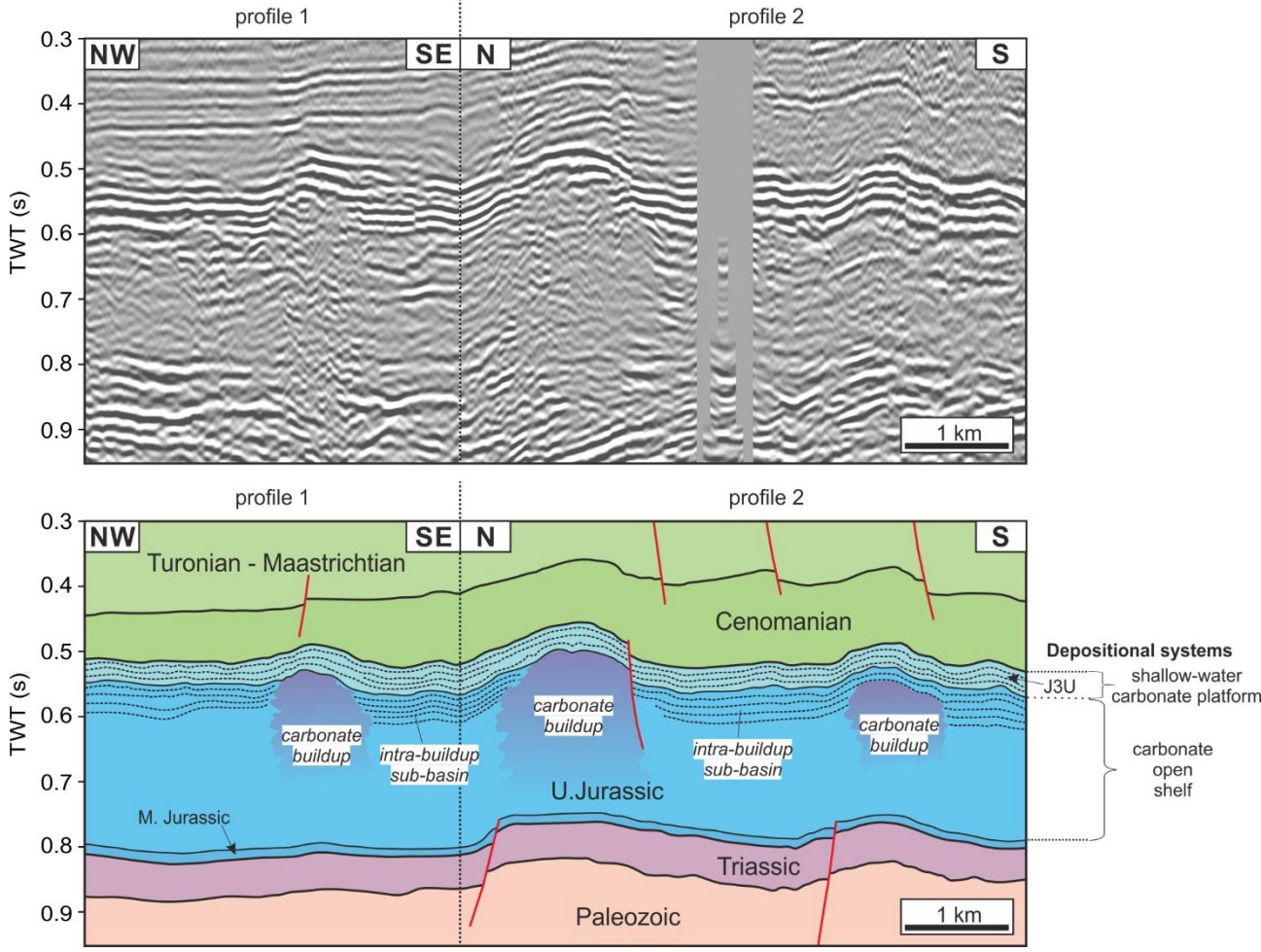

965

**Figure 15.** Uninterpreted and interpreted seismic transect (see Figure 5 for location) showing distinctive elements of depositional architecture of the Upper Jurassic succession in the study area: presence of large carbonate buildup complexes represented by massive facies and intra-buildup sub-basins, represented by bedded facies.