# Peer review of "Upper Jurassic carbonate buildups in the Miechów Trough, Southern Poland – insights from seismic data interpretation"

_Solid Earth, 2019_

## Referee Comment (RC1) · Jacek Matyszkiewicz (Referee) · 17 Feb 2020

The article is a valuable paper, contains new data and interesting interpretations, and is one of the first attempts of interpretation of seismic profiles in the Miechów Through and adjacent areas. However, several corrections and additions concerning mainly terminology seem necessary. Some key references have also been omitted.

Terminology The authors use the term "sponge megafacies" (lines 105, 153-158, 383) according to Matyja & Wierzbowski (2006). The widespread appearance of calcified

siliceous sponges in all Upper Jurassic successions of the northern Tethyan shelf commonly leads to the opinion that these organisms were the principal rock-forming components. Consequently, all these diversified facies are categorized into the far simplified term "sponge megafacies" (Matyja, 1976 fide Trammer, 1982; Matyja & Pisera, 1991). As the principal rock-forming components of these rocks are microbial structures (what Gwinner, 1971 has already pointed out) the term "microbial-sponge facies" or even "microbial facies" seems to be more adequate.

Literature In 2019, the PhD of A. Urbaniec was defended. It is an admittedly unpublished work, but the second author (PK) was reviewer. The dissertation concerns identical issues of seismic data interpretation in the Carpathian Foredeep. This work must be quoted and discussed. The authors conclude their remarks on stratigraphy on works from 2007-2009 (lines 136-137; 153-158). There is no basic work here of Olszewska et al. (2012) containing a critical analysis of previous work. This is a necessary item for quotation and brief discussion. The paper has not included issues related to differential compaction, although the authors devote a lot of space to it (lines 318-325; 380-381). This applies to publications Kochman & Matyszkiewicz (2013) - mechanical compaction and Matyszkiewicz & Kochman (2016) - chemical compaction. However, other works are cited (Matyszkiewicz et al., 2006, 2016 - line 381), in which only short paragraphs are devoted to the compaction.

Figures Figs. 8-15. In the lower parts the figures contain interpretations. This is not an interpretation from geological point of view because the vertical scale is given in seconds and not in meters. The interpreted seismic profiles should contain the vertical scale in meters. At least an additional explanation of the authors is required here. Fig. 16. In my opinion comparing of the wall of "Młynka" quarry (about 20 meters wide) with seismic profile with a length of about 5 km is inappropriate. Such a procedure can prove everything and negate everything.

List of additional references Gwinner, M.P. (1971). Carbonate rocks ofthe Upper Jurassic in SW-Germany. In: Müller, G. (ed.), Sedimentology of parts of Central Europe.

Kramer, Frankfurt a. M., pp. 193-207. Olszewska, B., Matyszkiewicz, J., Król, K. & Krajewski, M. (2012). Correlation of the Upper Jurassic-Cretaceous epicontinental sediments in southern Poland and south western Ukraine based on thin section. Biuletyn Państwowego Instytutu Geologicznego, 453: 29–80. Kochman, A. & Matyszkiewicz, J. (2013). Experimental method for estimation of compaction in the Oxfordian bedded limestones of the southern Kraków-CzÄŹstochowa Upland, Southern Poland. Acta Geologica Polonica, 63: 681-696. Matyszkiewicz, J. & Kochman, A. (2016). Pressure dissolution features in Oxfordian microbial-sponge buildups with pseudonodular texture, Kraków Upland, Poland. Annales Societatis Geologorum Poloniae, 86/4: 355–377. Urbaniec, A. (2019). Lithofacial development of the Upper Jurassic and Lower Cretaceous deposits in the DÄĚbrowa Tarnowska-DÄŹbica area based on the 3D seismic interpretations. Unpublished PhD Thesis. Faculty of Geology, Geophysics and Environmental Protection of AGH, Kraków.
* * *

---

## Referee Comment (RC2) · Tadeusz Peryt (Referee) · 4 Mar 2020

[revised manuscript text omitted]
400 structures from the study area but general depositional features are comparable with those observed on seismic data. Relationships between the massive facies, representing carbonate buildup deposits, and the bedded facies forming intra-buildup sub-basin observed on the field example (Fig. 16a and c), are also visible on seismic profiles shown in Fig. 16b and d. However, it should be stressed that to some degree field interpretations presented in Fig. 16a and c might be, due to vegetation and slope processes hindering visibility of outcrop, ambiguous and should be regarded as tentative; this includes (1) border between the
405 massive and basinal facies, and (2) the exact lateral and vertical extent of the intra-buildup sub-basin as well as the intra-basinal stratification. Tentative elements of outcrop interpretation were shown using dotted lines in Fig. 16a and c. In the outcrop, onlapping bedded facies are visible that partly overlie top of the massive facies (Fig. 16a). They can also be observed on seismic data (Fig. 16b). Upper Jurassic depositional architecture from the Kraków-Częstochowa Upland, expressed by the presence of carbonate buildups separated by intra-buildup sub-basins, can be clearly observed in outcrop example from Fig.
410 16c; its seismic-scale equivalent from the study area is shown in Fig. 16d. Comparison of field and seismic 
[revised manuscript text omitted]

[Figure]

**Figure 16.** Field examples from Młynka Quarry (Kraków-Częstochowa Upland, see Figure 3 for location) showing geometric relationship between the Upper Jurassic massive facies (carbonate buildups - their key contours are marked with red lines) and the bedded facies (yellow lines). Dotted red and yellow lines mark the ambiguous, partly tentative elements of outcrop interpretation. The outcrops examples (**a, c**) are compared to their seismic-scale equivalents from the Miechów Trough (**b, d**). Seismic example (**b**) is part of the profile from Fig. 11, seismic example from (**d**) is part of the profile from Fig. 15. Onlap contacts are marked with yellow arrows.

---

## Short Comment (SC1) · 11 Mar 2020

This is an intriguing and thought-provoking paper in the sense that it deals with an "evergreen" subject for those of us who work in the petroleum industry, i.e. the seismic indentification of reservoir-grade carbonate build-ups. The comments below should help to produce a final version: 1) On figure 2, the analogue areas mentioned should be annotated and a few more relevant examples should be added, "closer to home", i.e. in Austria, Czech Republic and Poland. Adámek, J., 2005. The Jurassic floor of the Bohemian Massif in Moravia–geology and paleogeography. Bulletin of Geosciences,

80(4), pp.291-305.

Zimmer W., Wessely G. (1996): Exploration results in thrust and subthrust complexes in the Alps and below the Vienna Basin in Austria. In: Wessely G., Liebl W. (eds) Oil and gas in Alpidic Thrustbelts and Basins of Central and Eastern Europe. EAGE Special Publication 5, Geological Society, London, 81–107. Wessely G. (2006): Geologie von Niederösterreich. Geologische Bundesanstalt, Wien. Mys′liwiec, Michal, Zenon Borys, Beata Bosak, Bogusław Liszka, Kazimierz Madej, Andrzej Maksym, Krystyna Oleszkiewicz, Małgorzata Pietrusiak, BozËŹena Plezia, Grzegorz Staryszak, GrazËŹyna S′ wieËŻ tnicka, Czesława Zielin′ ska, Krystyna Zychowicz, Piotr Gliniak, Radosław Florek, Jarosław Zacharski, Andrzej Urbaniec, Adam Go′ rka, Piotr Karnkowski, and Paweł H. Karnkowski, 2006, Hydrocarbon resources of the Polish Carpathian Foredeep: Reservoirs, traps, and selected hydrocarbon fields, in J. Golonka and F. J. Picha, eds., The Carpathians and their foreland: Geology and hydrocarbon resources: AAPG Memoir 84, p. 351 – 393.

2) I would certainly include the reference to this paper and also paint the position of the Polish Upper Jurassic reefs in a global context, such as reef types and reef builders: Wolfgang Kiessling, Erik Flügel and Jan Golonka (1999) Paleoreef Maps: Evaluation of a Comprehensive Database on Phanerozoic Reefs. AAPG Bulletin, 83, 1552–1587.

3) Frankly, on some of thee seismic sections, the detection limit for the interpretation is a challenge. It would mbe good to provide some close-ups on some of the features, e.g. on Figure 9, the singular carbonate build-up. . . I wonder whether some other seismic displays, such as inst. frequency or interval velocity, may be more helpful to show the presence and outline of these build-ups in a more convincing manner? Any sensitivity work on the potential use of velocity pull-up, i.e. could one expect to see one at all, or all these carbonates have pretty much the same velocity, i.e. variations less than, say, 5-10%?

4) Figure 16 is an interesting attempt to compare "apples and oranges", but I would not

do it. Regardless of the order of magnitude difference in scales, the outcrop photos are just not that convincing to see the difference between the massive and bedded facies. I suggest to drop this figure.

———————————————————

---

## Referee Comment (RC3) · Gabor Tari (Referee) · 25 Mar 2020

This is an intriguing and thought-provoking paper in the sense that it deals with an "evergreen" subject for those of us who work in the petroleum industry, i.e. the seismic identification of reservoir-grade carbonate build-ups. The comments below should help to produce a final version:

1) On figure 2, the analogue areas mentioned should be annotated and a few more relevant examples should be added, "closer to home", i.e. in Austria, Czech Republic
and Poland. Adámek, J., 2005. The Jurassic floor of the Bohemian Massif in Moravia–geology and paleogeography. Bulletin of Geosciences 80(4), pp.291-305. Zimmer W., Wessely G. (1996): Exploration results in thrust and subthrust comï£¿plexes in the Alps and below the Vienna Basin in Austria. In: Wessely G., Liebl W. (eds) Oil and gas in Alpidic Thrustbelts and Basins of Central and Eastern Euï£¿rope. EAGE Special Publication 5, Geological Society, London, 81–107. Wessely G. (2006): Geologie von Niederösterreich. Geologische Bundesanstalt, Wien. Mysliwiec, Michal, Zenon Borys, Beata Bosak, Bogusław Liszka, Kazimierz Madej, Andrzej Maksym, Krystyna Oleszkiewicz, Małgorzata Pietrusiak, BozZena Plezia,Grzegorz Staryszak, GrazËZyna SwieËZ tnicka, Czesława Zielinska, Krystyna Zychowicz, Piotr Gliniak, Radosław Florek, Jarosław Zacharski, Andrzej Urbaniec, Adam Gorka, Piotr Karnkowski, and Paweł H. Karnkowski, 2006, Hydrocarbon resources of the Polish Carpathian Foredeep: Reservoirs, traps, and selected hydrocarbon fields, in J. Golonka and F. J. Picha, eds., The Carpathians and their foreland: Geology and hydrocarbon resources: AAPG Memoir 84, p. 351 – 393.

2) I would certainly include the reference to this paper and also paint the position of the Polish Upper Jurassic reefs in a global context, such as reef types and reef builders: Wolfgang Kiessling, Erik Flügel and Jan Golonka (1999) Paleoreef Maps: Evaluation of a Comprehensive Database on Phanerozoic Reefs. AAPG Bulletin, 83, 1552–1587.

3) Frankly, on some of thee seismic sections, the detection limit for the interpretation is a challenge. It would be good to provide some close-ups on some of the features, e.g. on Figure 9, the singular carbonate build-up. . . I wonder whether some other seismic attribute displays, such as inst. frequency or interval velocity, may be more helpful to show the presence and outline of these build-ups in a more convincing manner? Any sensitivity work on the potential use of velocity pull-up, i.e. could one expect to see one at all, or all these carbonates have pretty much the same velocity, i.e. variations less than, say, 5-10%? Any analogue studies in this regard? How about those stunning Miocene examples from the Far East, Natuna, etc.? I am sure that some of those reefs
could provide some analogues.

4) Figure 16 is an interesting attempt to compare "apples and oranges", but I would not do it. Regardless of the order of magnitude difference in scales, the outcrop photos are just not that convincing to see the difference between the massive and bedded facies. I suggest to drop this figure.

---

## Author Response (AR1)

Dear Elias,

Please find our responses to reviewers' comments and the revised paper and figures of our manuscript "Upper Jurassic carbonate buildups in the Miechów Trough, Southern Poland – insights from seismic data interpretation".

I would like to first address our reply to the reviewers' comments.

Our reply to Prof. Jacek Matyszkiewicz:

We would like to thank Jacek Matyszkiewicz for his valuable and constructive comments, they surely helped to finally shape our paper. Please find below response to all the issues raised in your review.

**Comment from referee (RC1):**

*RC1: "Terminology - The authors use the term "sponge megafacies" (lines 105, 153-158, 383) according to Matyja & Wierzbowski (2006). The widespread appearance of calcified siliceous sponges in all Upper Jurassic successions of the northern Tethyan shelf commonly leads to the opinion that these organisms were the principal rock-forming components. Consequently, all these diversified facies are categorized into the far simplified term "sponge megafacies" (Matyja, 1976 fide Trammer, 1982; Matyja & Pisera, 1991). As the principal rock-forming components of these rocks are microbial structures (what Gwinner, 1971 has already pointed out) the term "microbial-sponge facies" or even "microbial facies" seems to be more adequate."*

**Authors response:**

We used term "sponge megafacies" in the description of the general geological background for our results that are based on interpretation of seismic data. This term was derived from the literature as our data are of course of absolutely different resolution and do not allow for discriminating, directly or indirectly, any rock-forming components. Our intention was to treat term "sponge megafacies" as a general term, coined in the literature, with certain stratigraphic connotations. However, we do understand and do agree that it is being used as general, partly informal descriptive term for the Upper Jurassic carbonate rocks. Our understanding is that this term does not automatically imply that sponges were the principal rock-forming component, with microbial structures playing also very important role; therefore, we rephrased our text ([1]line 105) in order to clearly emphasize that carbonate buildups deposits are built of sponges and microbialites. Detailed discussion regarding intricacies of local versus regional stratigraphy, primary and secondary rock constituents etc. of the Upper Jurassic succession should be had between specialists working with appropriate data, and
* * *
[1] we are referring to line numbers of the original manuscript

having appropriate know-how and experience. Seismic data could provide very interesting, sometime novel insight regarding various aspects of structure and evolution of this carbonate succession but such problems are clearly beyond its reach.

**Comment from referee (RC1):**

RC1: *"Literature - In 2019, the PhD of A. Urbaniec was defended. It is an admittedly unpublished work, but the second author (PK) was reviewer. The dissertation concerns identical issues of seismic data interpretation in the Carpathian Foredeep. This work must be quoted and discussed."*

**Authors response:**

In the submitted version of the paper, we followed widely accepted and adhered to by many journals rule that unpublished studies, including PhD theses, should not be cited. In our paper we made just one exception – we cited unpublished well reports for wells used to calibrate seismic data from our study area, as they contain crucial information absolutely necessary to properly illustrate various aspects of seismic data interpretation. Archive industry reports in most cases remain indefinitely unpublished hence our decision. On the other hand, we do not have any problems with citing this particular PhD thesis as it certainly is relevant to our results. We have consulted this with Solid Earth editors, and, following their approval, remarks on this unpublished work was added to our paper (lines 53 and 367). It should be also stressed that this study is based on 3D seismic data (we used a bit more regional 2D seismic coverage) from different part of the basin with partly different geological history, and does not provide any crucial information that would in any way alter our own results.

**Comment from referee (RC1):**

RC1: *"There is no basic work here of Olszewska et al. (2012) containing a critical analysis of previous work. This is a necessary item for quotation and brief discussion."*

**Authors response:**

Upper Jurassic succession in S Poland, similarly to rest of the Europe, has been intensively studied for more than 200 years. This certainly resulted in publication of huge number of various papers dealing with very different aspects of Jurassic stratigraphy etc. Over last couple of decades various opinions have been formulated in this context, and, as a result, we faced very complex task of selecting key papers that would best illustrate such diversity of opinions. In the process we surely we might have missed some papers that in other's eyes are very important. Therefore, without any hesitation, we followed advice of Jacek Matyszkiewicz and added Olszewska et al., 2012 to the references (lines 137 and 156). We would like to stress that discussion of very detailed Upper Jurassic stratigraphy was outside scope of our work, simply due to lack of adequately detailed well data from our study area. We had access to old wells with stratigraphy available in archive well reports based on divisions from

many decades ago, and to two more recently drilled wells in which however no detailed stratigraphic studies have been performed. Therefore, we were forced to use rather simplified stratigraphic subdivisions, heavily relying on vertical lithological variations derived from well logs and rock cuttings. Hopefully, future stratigraphic studies will more fully clarify Jurassic stratigraphy and this knowledge could be used in future seismostratigraphic studies. Our conclusions of generic character would remain largely unchanged, only stratigraphic context might be partly different.

**Comment from referee (RC1):**

*RC1: "The paper has not included issues related to differential compaction, although the authors devote a lot of space to it (lines 318-325; 380-381). This applies to publications Kochman & Matyszkiewicz (2013) – mechanical compaction and Matyszkiewicz & Kochman (2016) - chemical compaction. However, other works are cited (Matyszkiewicz et al., 2006, 2016 - line 381), in which only short paragraphs are devoted to the compaction."*

**Authors response:**

This is a problem partly similar to the problem with selection of papers devoted to Jurassic stratigraphy - we tried to select the best and most-to-the-point papers dealing with compaction of Jurassic carbonates but certainly we might have missed some of them. Following this suggestion, our reference list was supplemented.

**Comment from referee (RC1):**

*RC1: "Figures - Figs. 8-15. In the lower parts the figures contain interpretations. This is not an interpretation from geological point of view because the vertical scale is given in seconds and not in meters. The interpreted seismic profiles should contain the vertical scale in meters. At least an additional explanation of the authors is required here."*

**Authors response:**

It really depends on what one defines as "geological interpretation of seismic data". There are hundreds if not thousands of papers that contain interpreted seismic data in time domain, with vertical scale given in seconds of two-way travel time. Nowadays time seismic data still prevail although of course more and more frequently also depth data is available due to wider application of processing techniques such as PSDM etc. In this case however only time data was available so the only option was to present uninterpreted profiles and their interpreted equivalents in vertical time scale. This is standard approach that could be illustrated by a very large number of papers based on seismic data, and for us this is geological interpretation of seismic data, indeed. It should be also stressed that our interpreted data is not entirely depth-independent. Detailed time-depth relationships are clearly given on Figures 6 and 7 (cf. also Figures 11 and 12), and this information should be sufficient to properly

asses an overall geometry of the studied carbonate buildups etc. and put our time interpretation in depth context.

**Comment from referee (RC1):**

RC1: *"Fig. 16. In my opinion comparing of the wall of "Młynka" quarry (about 20 meters wide) with seismic profile with a length of about 5 km is inappropriate. Such a procedure can prove everything and negate everything."*

**Authors response:**

Indeed, maybe this was a bit too long shot. All we wanted to achieve here was to show that some geometrical relationship between bedded and massive facies, although in different scales, could be observed both in outcrops and on seismic data. However, we agree that these might be different features, so detailed comparative study of outcrops and seismic profiles might require additional field work, possibly combined with seismic stratigraphic modelling studies, similar to the work of W. Schlager et al. in the Dolomites. Taking this into account, and also the fact that similar concerns have been raised by another Reviewer, we decided to remove this comparison from this paper. Accordingly, we removed relevant parts from the manuscript, i.e. lines 68–71 (Introduction), and lines 396–410 (Discussion). This change did not however substantially influence any element of our interpretation and they all still hold valid.

**Comment from referee (RC1):**

RC1: *"List of additional references (…)."*

**Authors response:**

All suggested references were added to the reference list (for detailed information please see the supplementary file).

**Authors changes in manuscript**:

Please find attached the supplement file with listed specific changes in the manuscript.

We would like to thank again for all the comments and suggestions, they significantly helped us to refine our paper.

Łukasz Słonka

(on behalf of the authors)

**Authors response to Reviewer 1 (RC1):**

**Supplement (detailed list of corrections)**

**Text:**

Line 53, additional sentence was added: Recently, several new buildups have been identified and interpreted using seismic data (Urbaniec, 2019). However those results represent more southern part of the basin, located beneath the Miocene of the Carpathian Foredeep Basin (about 50 km to the south from the study area).

Line 53, the next sentence was move to the new paragraph.

Lines 104-107, we have changed this part of the text according to the suggested corrections. Additional references were added (Gwinner, 1971). Reference to Fig.4a was additionally moved from line 107 to line 105. Modified part of the text is given below:

The Oxfordian and lower Kimmeridgian succession within the Polish part of the northern Tethyan shelf margin is commonly interpreted as a carbonate ramp or an open shelf deposits (e.g. Matyja et al., 1989; Kutek, 1994; Gutowski et al., 2005; Matyja, 2009; Krajewski et al., 2011; Fig. 4a). These deposits, sometimes termed the sponge megafacies, are bulit of sponges and microbialites, and are present within the entire European part of the northern Tethyan shelf margin (Gwinner, 1971; Matyja, 1977; Matyja and Pisera, 1991; Matyja and Wierzbowski, 1995, 1996, 2006; cf. Matyszkiewicz, 1997a; Gutowski et al., 2005, 2006).

Line 107, also, we added here additional reference (cf. Matyszkiewicz, 1997a)

Line 137, new reference was added (Olszewska et al. 2012)

Line 156, a new sentence related to the above reference (Olszewska et al., 2012) was added:

Further micropaleonotlogical investigations allowed for stratigraphical reassessment of the Upper Jurassic strata beneath the central part of the Carpathian Foredeep basin, as well as for the regional correlations towards the south-western Ukraine (Olszewska et al. 2012).

Line 367, we comment on the work by A. Urbaniec in the new sentence added to the text, which is given below:

Recently, Urbaniec (2019) provided seismic examples of Upper Jurassic carbonate buildups of similar size that are located about 50 km south-east from the study area. Those buildups are characterised by complex geometries and probably consist of several levels of the massive limestones.

Line 381, additional references devoted to various aspects of differential compaction were added to the text: (Kochman & Matyszkiewicz, 2013; Matyszkiewicz and Kochman, 2016)

**Figures:**

Figure 16, the entire figure was removed together with related part of the text in the Discussion (lines 396–410); also the appropriate sentence in the Introduction was skipped (lines 68–71).

Łukasz Słonka
(on behalf of the authors)

Our reply to Prof. Tadeusz Peryt:

We would like to thank Tadeusz Peryt for his valuable corrections of our manuscript. We will address below to your key comments/suggestions:

For lines 75 and 85 change from "Late Cretaceous" to "late Cretaceous" was suggested for the age of the inversion and related formation of the Szczecin-Łódź-Miechów Synclinorium. In our paper, we adopted terminology from the International Chronostratigraphic Chart (http://www.stratigraphy.org/index.php/ics-chart-timescale) and general rules provided by the International Commission on Stratigraphy, so, accordingly, series and epoch were written by uppercase. It has been well established in the literature that inversion of the Southern part of the Polish Basin embraced Turonian - Maastrichtian, so "Late" rather than "late" was used in our text and we still think that this was correct approach.

Figure 3 – we agree that the citation of maps should be chronological. In the Figure 3, only the position of the Northern boundary of the Miocene deposits of the Carpathian foredeep basin was used from the map of *Żytko et al., (1988*). The rest of this map is based on *Dadlez et al., (2000)* and is not based on the map by *Żytko et al., (1988*). Therefore, we decided to modify this map and to move reference to *Żytko et al., 1998* from the figure caption to the map legend.

Figure 4 – yes, we have missed "HST" here, it was added in the corrected version of this figure.

Finally, we followed and agreed with all other comments and suggested corrections., and we provided a specific list of all the changes made in manuscript in the additional file, as a supplement to our response.

Once again, thank you for all your suggestions and corrections. They greatly improved our paper and increased its quality.

Łukasz Słonka
(on behalf of the authors)

**Authors response to Reviewer 2 (RC2):**

**Supplement (detailed list of corrections)**

**Text:**

Line 114, corrected (semicolon was used instead of comma)

Line 149, corrected („Lower" was be changed to „Early")

Line 164, corrected („rather" was removed)

Lines 168–169, well names (Michałów-3, Węchadłów-1, Lipówka-1, Chopin-1, Belvedere-1) were skipped

Line 219, corrected (italic style changed to normal)

Line 222, corrected (italic style changed to normal)

Line 263, corrected (British spelling was used)

Lines 275–276, corrected (italic style changed to normal)

Line 319, corrected („rather" was changed to „more")

Line 330, corrected (British spelling was used)

Line 360, corrected (British spelling was used)

Line 368, corrected („implication" was changed to „application")

Line 370, corrected: change in text ("different" was changed to „various")

Line 381, corrected (comma)

Line 450, corrected (British spelling was used)

**References:**

Line 513, citation style was corrected (comma instead dot)

Line 550, corrected ("Państwowego" was removed)

Line 638, corrected (comma was used instead of dot)

Line 826, corrected ("in Polish with English summary" will be added)

**Figures and figure captions:**

Figure 3, descriptions in the legend (see left side) were corrected as follows: Keuper – corrected, Devonian (unseparated) – corrected

Figure caption: reference to *Żytko et al.,1988* was moved from the figure caption to map legend.

Figure 4, HST" was added in this figure

Łukasz Słonka

(on behalf of the authors)

Our reply to Dr. Gabor Tari:

We would like to thank Gabor Tari for his thorough review of our paper, it certainly helped to increase its clarity and, we hope, an overall quality. Please find below our response to your comments.

**Comment from referee (RC3):**

*RC3: "1) On figure 2, the analogue areas mentioned should be annotated and a few more relevant examples should be added, "closer to home", i.e. in Austria, Czech Republic and Poland.*

*Adámek, J., 2005. The Jurassic floor of the Bohemian Massif in Moravia–geology and paleogeography. Bulletin of Geosciences 80(4), pp.291-305.*

*Zimmer W., Wessely G. (1996): Exploration results in thrust and subthrust complexes in the Alps and below the Vienna Basin in Austria. In: Wessely G., Liebl W. (eds) Oil and gas in Alpidic Thrustbelts and Basins of Central and Eastern Europe. EAGE Special Publication 5, Geological Society, London, 81–107.*

*Wessely G. (2006): Geologie von Niederösterreich. Geologische Bundesanstalt, Wien.*

*Mysliwiec, Michal, Zenon Borys, Beata Bosak, Bogusław Liszka, Kazimierz Madej, Andrzej Maksym, Krystyna Oleszkiewicz, Małgorzata Pietrusiak, Bożena Plezia,Grzegorz Staryszak, Grażyna Świętnicka, Czesława Zielińska, Krystyna Zychowicz, Piotr Gliniak, Radosław Florek, Jarosław Zacharski, Andrzej Urbaniec, Adam Gorka, Piotr Karnkowski, and Paweł H. Karnkowski, 2006, Hydrocarbon resources of the Polish Carpathian Foredeep: Reservoirs, traps, and selected hydrocarbon fields, in J. Golonka and F. J. Picha, eds., The Carpathians and their foreland: Geology and hydrocarbon resources: AAPG Memoir 84, p. 351 – 393."*

**Authors response:**

In the submitted version of the paper we cited selected papers dealing with various aspects of seismic interpretation of the Upper Jurassic carbonate buildups in central-western Europe, but we might have missed some papers that should have been mentioned; we have supplemented our reference list and added suggested references (line 50). Amended list of other relevant seismic studies has been also reflected on map from the Figure 2.

**Comment from referee (RC3):**

RC3: *"2) I would certainly include the reference to this paper and also paint the position of the Polish Upper Jurassic reefs in a global context, such as reef types and reef builders:*
*Wolfgang Kiessling, Erik Flügel and Jan Golonka (1999) Paleoreef Maps: Evaluation of a Comprehensive Database on Phanerozoic Reefs. AAPG Bulletin, 83, 1552–1587."*

**Authors response:**

We agree that description of global context for the Upper Jurassic carbonate buildups would be useful, especially for international readers from outside of Europe. Accordingly, we modify chapter 2.2. (new text was added from line 112), and also, the above-mentioned reference (Kiessling et al., 1999) was cited in the revised version of the manuscript.

**Comment from referee (RC3):**

RC3: *"3) Frankly, on some of these seismic sections, the detection limit for the interpretation is a challenge. It would be good to provide some close-ups on some of the features, e.g. on Figure 9, the singular carbonate build-up. . . "*

**Authors response:**

This is a very helpful comment. In order to better visualize carbonate buildups we supplemented Figures 8, 9, and 10 with additional zooms of the carbonate buildups so their external geometries and relationship to the surrounding deposits could be better appreciated, also because these zooms are less vertically exaggerated.

**Comment from referee (RC3):**

RC3: *"I wonder whether some other seismic attribute displays, such as inst. frequency or interval velocity, may be more helpful to show the presence and outline of these build-ups in a more convincing manner?"*

**Authors response:**

We agree that seismic attribute displays would be helpful for a more detailed investigation of the outline of the buildups, this is however fairly important issue requiring (and deserving) lengthy treatment and we decided, even before submitting our paper to Solid Earth, to devote a separate paper to this problem. Having this topic included in this paper would require either rather condensed treatment of this problem or would significantly expand the length of the text, and we found both options unviable.

**Comment from referee (RC3):**

*RC3: "Any sensitivity work on the potential use of velocity pull-up, i.e. could one expect to see one at all, or all these carbonates have pretty much the same velocity, i.e. variations less than, say, 5-10%?"*

**Authors response:**

The seismic velocity considerably differs between the massive and bedded facies. Interval velocity of the massive carbonates is about 5000–5500 m/s as suggested by modern Chopin-1 and Belvedere-1 wells, while bedded facies are characterized by lower values, in order of 3800 – 5000 m/s as proved by older wells. This may suggest that lateral seismic velocity variations between massive and bedded facies sometimes might exceed 10%, and such differences might be responsible for producing velocity pull-up effect that might be observed beneath the carbonate buildups. We'd like to stress however that reliable, good quality velocity data is available only for the massive facies as there are no modern wells drilled within the basinal bedded facies, so information on lateral velocity variations is rather sparse and qualitative. Nevertheless, we agree that some additional comments regarding lateral velocity variations, velocity pull-up effects etc. would be a welcome addition so we added paragraph on that to the corrected version of out manuscript. The new text starts from line 326.

**Comment from referee (RC3):**

*RC3: "Any analogue studies in this regard? How about those stunning Miocene examples from the Far East, Natuna, etc.? I am sure that some of those reefs could provide some analogues."*

**Authors response:**

Yes, we agree, and a sentence concerning analogous studies was added (it is included in the new paragraph that starts from line 326). We refer to the large Miocene carbonate buildups from the Far East (Luconia) where similar velocity pull-up effects were observed.

**Comment from referee (RC3):**

*RC3: "4) Figure 16 is an interesting attempt to compare "apples and oranges", but I would not do it. Regardless of the order of magnitude difference in scales, the outcrop photos are just not that convincing to see the difference between the massive and bedded facies. I suggest to drop this figure."*

**Authors response:**

With little hesitation we agree with this suggestion, perhaps comparison of outcrop and seismic profile presented in Figure 16 was a little bit to far reaching. By presenting this comparison we wanted to stress that some geometrical relationship between the massive and bedded carbonate facies, although in different scales, could be observed both in outcrops and on seismic data. However, we agree that our attempts to compare the geometrical relationship between main facies observed in outcrops and on seismic data might be not be fully convincing due to the too large differences in scales. Taking this into account, and also due to the fact that similar concerns have been raised by another Reviewer, we

decided to remove this comparison from our paper. Similarly, we removed also all corresponding parts in the text including lines 68–71 (Introduction) and lines 396–410 (Discussion). These changes did not, however, influence any of the key elements of our interpretation and they all still hold fully valid.

**Authors changes in manuscript**:

List of specific changes in our manuscript has been attached as the supplement file.

Again, many thanks for the comments and suggestions, they all significantly helped us to improve our paper.

Łukasz Słonka

(on behalf of the authors)

**Authors response to Reviewer 3 (RC3):**

**Supplement (detailed list of corrections)**

**Text:**

Lines 50, 53, following your suggestions we have added additional references: Zimmer and Wessely, 1996; Adámek, 2005; Wessely, 2006; Myśliwiec et al., 2006

Line 112, we have supplemented this part of the text, according to your corrections:

The Upper Jurassic carbonate buildups in southern Poland display a large diversity of reef types, from siliceous sponge mounds to microbial-sponge buildups and coral reefs, as all of these types were commonly found in Europe where reefs were most widespread in the Late Jurassic (Kiessling et al., 1999; cf. Leinfelder et al., 1996; Gliniak et al., 2005; Matyszkiewicz et al., 2012; Krajewski et al., 2018). Outside of Europe reefs occurred less commonly in Late Jurassic, and they represented mainly coral-dominated reefs and biostromes (Kiessling et al., 1999). Common carbonate buildup types that can be recognized from the seismic data in Poland are bioherms (e.g. Gliniak and Urbaniec, 2001, 2005; Gliniak et al., 2005). Worldwide, these organic structures can be found in all latitudes between 45°S and 52°N (Kiessling et al., 1999); in southern Poland they often developed as large microbial-sponge biohermal complexes (e.g. Matyja and Wierzbowski, 2006).

Line 326, we have added here new paragraph, devoted to velocity pull-up effect observed on the analyzed seismic data. Additionally, we made here a brief comment on the studies from Luconia,

Malaysia and NW shelf of Australia where the problem of velocity pull-ups was also raised (appropriated citations were added), as suggested. The entire new text is shown below:

The velocity pull-up effect observed beneath the carbonate buildups (cf. Fig. 13) results from lateral seismic velocity contracts between the massive and stratified (bedded) carbonates. The interval velocity of the massive limestones, drilled by modern Chopin-1 and Belvedere-1 wells, is about 5000–5000 m/s and is significantly higher in comparison to seismic velocity obtained from the Michałów-3 or Lipówka-1 legacy wells for the corresponding stratified deposits that are in order of ca. 3800-5000 m/s. However, it should be taken into account that velocity information from these old wells should be treated only tentatively, due to their uncertainty resulting from the lower quality of older well-logging data. Expected lateral seismic velocity variations between the massive and bedded carbonates often exceed 10% and might be responsible for producing some velocity pull-ups beneath the seismically faster carbonate buildups. Then, it is probable that at least for some of the morphological heights situated beneath the carbonate buildups in the analysed time seismic data, velocity pull-ups might have distorted their true geometries. The similar role of high-velocity reefal intervals in production of velocity pull-up effects beneath the carbonate buildups was described for time seismic data characterising the large Miocene buildups in Luconia, Malaysia (e.g. Zampetti et al., 2004; Rankey et al. 2019) or numerous isolated buildups from the north-west shelf of Australia (Saquab and Bourget, 2016).

**Figures and figure captions:**

Figure 2, according to the suggested correction, we have annotated (by subsequent numbers 1, 2, 3 etc.) the location of previous seismic interpretation studies/papers on the Upper Jurassic carbonate buildups and we added the particular references in the figure caption, including suggested citations from Austria and Czech Republic.

[Figure]

**Figure 2.** Simplified paleogeographic sketch map of central and western Europe for the middle–late Oxfordian (after Wierzbowski et al., 2016); red points show location of the previously published seismic interpretation studies/papers dealing with the Upper Jurassic carbonate buildups from the northern Tethyan shelf margin and adjacent areas (1. Ellis et al., 1990; 2. Bunes et al., 2010; 3. Hartmann et al., 2012; 4. Lüschen et al., 2014; 5. Zimmer and Wessely, 1996; 6. Adámek, 2005; 7. Gliniak and Urbaniec, 2001; 8. Gliniak et al., 2005; see text for more details).

Figures 8–10, for each figure we added zoom of identified carbonate buildups; also, the figure captions have been modified accordingly.

[Figure]

**Figure 8. (a)** Uninterpreted and interpreted seismic profile (12-5-92K) from the Miechów Trough, see Figure 5 for location. Major NW-SE oriented Opatkowice and Kostki Małe fault zones are rooted in the Paleozoic basement and associated with inversion anticlines developed within the Mesozoic cover; **(b)** Two carbonate buildups were identified in this profile; one of them was partly drilled by the SLE Chopin-1 well.

[Figure]

**Figure 9. (a)** Uninterpreted and interpreted seismic profile (11-5-92K) from the Miechów Trough, see Figure 5 for location. Major NW-SE oriented Opatkowice and Kostki Małe fault zones are rooted in the Paleozoic basement and associated with inversion anticlines developed within the Mesozoic cover; **(b)** One isolated carbonate buildup was identified in this profile.

[Figure]

**Figure 10. (a)** Uninterpreted and interpreted seismic profile (10-5-92K) from the Miechów Trough, see Figure 5 for location. Major NW-SE oriented Opatkowice and Kostki Małe fault zones are rooted in the Paleozoic basement and associated with inversion anticlines developed within the Mesozoic cover; **(b)** Two carbonate buildups were identified in this profile.

Figure 16, we dropped this figure entirely together with the related part of the text in the Discussion (lines 396–410). Also, we removed the appropriate sentence in the Introduction which concerned this part of our analysis (lines 68–71), as these considerations will be no longer element of this paper. Due to fact that the outcrop examples from Figure 16 were removed from the text, we also removed location of the Młynka Quarry from Figure 3.

Łukasz Słonka
(on behalf of the authors)

Again, we would like express our gratitudes to the Reviewers for their comprehensive reviews of our paper. All of their comments have been addressed above and with revision to the text and the figures. Additionally during our work on revised version of this paper, we have found some small, technical errors in our original manuscript. We decided to correct all of them; detailed list of these additional corrections can be found in the attached supplement file (see below). Finally, we attached the marked-up manuscript version (track changes in Word) converted into a pdf, which includes all our changes and corrections.

I hope that the revised version of our manuscript proves acceptable for publication in the Solid Earth.

Sincerely,

Łukasz Słonka
14th May 2020

**Supplement file for the Editor**

**List of additional changes/corrections in the manuscript, done by authors, with explanations:**

**Text:**

Line 52, in order to provide full panorama of previous seismic studies of Upper Jurassic carbonate succession in S Poland we decided to expand and amend reference list include also those publications that have been published in local journals, only in Polish. The following citations were added to the reference list: Gliniak et al., 2000, 2001; Gliniak and Urbaniec, 2001; Misiarz et al., 2004; Jędrzejowska-Tyczkowska et al., 2005.

Line 52, we have replaced also the word "all" by "usually".

Line 83, in 1997, A. Feldman-Olszewska published two works, the corrected citations is as follows:

Feldman-Olszewska, A.: Depositional systems and cyclicity in the intracratonic Early Jurassic basin in Poland, Geol. Q., 41(4), 475–489, 1997a.
Feldman-Olszewska, A.: Depositional architecture of the Polish epicontinental Middle Jurassic basin, Geol. Q., 41(4), 491–508, 1997b.

In the corrected version (line 83), we changed "Feldman-Olszewska, 1997" to "Feldman-Olszewska, 1997a, 1997b".

Line 102, citation "Gutowski et al., 2005" was moved in order to keep chronological order of citations.

Line 104, „deposit" was changed to „deposits"

Lines 105 and 107, we moved the reference to Figure 4a in text from line 107 to line 105. This correction partly results from the modification of this entire paragraph that was done following suggestions from one of the Reviewers.

Lines 477, 548, 561, 583, 612, 660, 664, 674, 707, 723, for references in the revised manuscript we have removed additional numbers for the volumes. This was done by us in order to correct and unify the reference style, because these additional number of volumes were unnecessarily placed by us in the original manuscript.

**Figures and figure captions:**

Figure 4,

- term „sponge megafacies" was skipped in legend (right side of the figure).
- text corrections in figure caption (British spelling will be used; "after" was deleted)

Łukasz Słonka

(on behalf of the authors)

*we are referring to line numbers of the original manuscript version

[revised manuscript text omitted]
, approximately 20 km north-west from Kraków (Fig. 3; e.g. Matyszkiewicz, 1993, 1997b). The size of the Upper Jurassic carbonate buildups observed in the Młynka Quarry (with observed heights about 15–20 m), is smaller than seismic-scale structures from the study area but general depositional features are comparable with those observed on seismic data.~~

435 ~~Relationships between the massive facies, representing carbonate buildup deposits, and the bedded facies forming intra-buildup sub-basin observed on the field example (Fig. 16a and c), are also visible on seismic profiles shown in Fig. 16b and d. However, it should be stressed that to some degree field interpretations presented in Fig. 16a and c might be, due to vegetation and slope processes hindering visibility of outcrop, ambiguous and should be regarded as tentative; this includes (1) border between the massive and basinal facies, and (2) the exact lateral and vertical extent of the intra-buildup sub-basin as well as the intra-~~

440 ~~basinal stratification. Tentative elements of outcrop interpretation were shown using dotted lines in Fig. 16a and c. In the outcrop, onlapping bedded facies are visible that partly overlie top of the massive facies (Fig. 16a). They can also be observed on seismic data (Fig. 16b). Upper Jurassic depositional architecture from the Kraków-Częstochowa Upland, expressed by the presence of carbonate buildups separated by intra-buildup sub-basins, can be clearly observed in outcrop example from Fig. 16c; its seismic-scale equivalent from the study area is shown in Fig. 16d. Comparison of field and seismic~~
[revised manuscript text omitted]

[Figure]

**Figure 16.** Field examples from Młynka Quarry (Kraków-Częstochowa Upland, see Figure 3 for location) showing geometric relationship between the Upper Jurassic massive facies (carbonate buildups - their key contours are marked with red lines) and the bedded facies (yellow lines). Dotted red and yellow lines mark the ambiguous, partly tentative elements of outcrop interpretation. The outcrops examples (**a, c**) are compared to their seismic-scale equivalents from the Miechów Trough (**b, d**). Seismic example (**b**) is part of the profile from Fig. 11, seismic example from (**d**) is part of the profile from Fig. 15. Onlap contacts are marked with yellow arrows.

995